# RA-VLA: Retrieval-Augmented VLA for Test-Time Adaptation

**Sanghwan Jang**[1]  **Minjin Jeon**[2]  **Minsoo Kim**[2]  **Seongjin Choi**[1]  **Dongha Kim**[1]  **Hwanjo Yu**[1 2]

## Abstract

Vision-Language-Action (VLA) models provide a versatile foundation for general robotic manipulation, yet they exhibit significant brittleness when confronted with novel task distributions. While In-Context Imitation Learning (ICIL) offers a training-free alternative, existing frameworks suffer from an *adaptation bottleneck* that hinders the effective translation of expert context to executable actions. This failure originates from superficial retrieval mechanisms and an inherent *behavioral inertia* that anchors the policy to its pre-trained priors. To address these limitations, we present RA-VLA, a retrieval-augmented VLA framework that integrates behavior-aligned context retrieval with a grounded execution pipeline. By enforcing faithful adherence to functional cues within a scalable architecture, RA-VLA facilitates seamless task adaptation while preserving inference efficiency. Our empirical evaluations across the LIBERO benchmark and a real-world UR5e environment demonstrate that RA-VLA achieves superior success rates and computational efficiency, establishing a robust framework for training-free robotic adaptation.

## 1. Introduction

Vision-Language-Action (VLA) models (Kim et al., 2025b; Pertsch et al., 2025; Li et al., 2024; Kim et al., 2025a; Black et al., 2025b;a; Bjorck et al., 2025) have emerged as a promising paradigm for general-purpose robotic manipulation, harnessing the vast knowledge embedded in multimodal foundation models (Karamcheti et al., 2024; Steiner et al., 2024; Chen et al., 2025) and large-scale robotic datasets (Walke et al., 2023; Khazatsky et al., 2024; O'Neill et al., 2024). Despite their impressive capabilities, existing

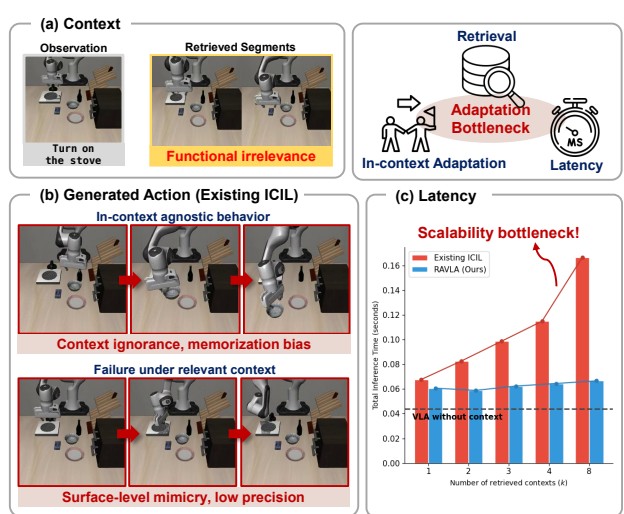

Figure 1. ***Adaptation bottleneck*** **in existing ICIL frameworks.** Existing ICIL methods (a) prioritize superficial visual similarity over functional intent in context retrieval, (b) fail to effectively leverage contextual guidance, and (c) exhibit significant computational overhead for larger contexts. While this overhead is plotted for RICL, other ICIL baselines follow a nearly identical trend.

VLAs exhibit significant brittleness when confronted with unseen manipulation tasks (Guruprasad et al., 2024; Fang et al., 2025; Fei et al., 2025; Gu et al., 2025). Specifically, when tasked with novel instructions, they predominantly revert to familiar behaviors from their training distribution or produce erratic motions that fail to align with the intended goal. This inherent reliance on static pre-trained knowledge poses a challenge for seamless deployment; effectively adapting high-capacity VLAs to new tasks necessitates an expensive and non-scalable re-training process to remain functional in dynamic environments.

To overcome this limitation, In-Context Imitation Learning (ICIL) (Fu et al., 2025; Yoo et al., 2025; Zhu et al., 2024; Sridhar et al., 2025) has recently gained attention as a viable means of achieving test-time adaptation. This approach bypasses the expensive re-training process by providing a few expert demonstrations as in-context cues, which steer the policy toward new task objectives without explicit weight updates. By directly incorporating expert demonstrations into the input prompt, these frameworks attempt to provide behavioral guidance to reconcile the model's pre-trained knowledge with unseen task demands.

[1]Department of Computer Science and Engineering, POSTECH, Pohang, South Korea [2]Graduate School of Artificial Intelligence, POSTECH, Pohang, South Korea. Correspondence to: Hwanjo Yu <hwanjoyu@postech.ac.kr>.

*Proceedings of the 43rd International Conference on Machine Learning*, Seoul, South Korea. PMLR 306, 2026. Copyright 2026 by the author(s).

Despite their promise, existing ICIL frameworks suffer from an ***adaptation bottleneck***—the inability to effectively translate in-context guidance into precise actions. This bottleneck is evidenced by the failure case of unseen *'turn on the stove'* task in Figure 1. As shown in (a), the current retriever prioritizes superficial visual similarity over functional intent, yielding behaviorally inconsistent guidance that misleads the policy. This issue is further exacerbated by *behavioral inertia* (b), where the policy fails to modulate its execution based on the provided in-context cues and remains anchored to its pre-trained *bowl-picking* prior. Even when attempting to directly utilize the retrieved actions, the policy fails to compensate for the discrepancies between the expert demonstration and the current scene. Furthermore, as illustrated in (c), current architectures incur significant computational overhead, with latency scaling poorly as the number of retrieved segments increases.

To bridge this gap, we propose RA-VLA, a retrieval-augmented VLA framework that synergizes action-aware retrieval with a context-grounded execution pipeline to facilitate seamless in-context adaptation. Specifically, we introduce *behavioral alignment loss* to train the retriever, mapping behaviorally similar expert segments close together in a shared latent space rather than relying on superficial visual features. The expert guidance retrieved from the aligned space ensures behavioral congruence with the target task demands, providing a robust foundation for in-context adaptation. Building on our retrieval mechanism, we introduce *contextual adherence loss* to break the behavioral inertia rooted in the policy's pre-trained priors. By enforcing an action regression margin between relevant and irrelevant expert segments, this loss incentivizes the policy to ground its action generation in the retrieved context. Crucially, these components are integrated into a scalable architecture that pre-encodes and caches each segment independently, thereby minimizing the computational overhead of incorporating expert guidance to the policy.

We evaluate the proposed framework across both simulation and real-world platforms, including the LIBERO benchmark (Liu et al., 2023) and a UR5e robot environment. By testing on tasks entirely excluded from training, we demonstrate that RA-VLA can successfully execute novel behaviors via in-context guidance without any weight updates. Specifically, RA-VLA outperforms existing state-of-the-art baselines, yielding absolute success rate improvements of 17.60% on the LIBERO benchmark and 20.83% in the UR5e environment. Notably, enabled by our decoupled encoding design, the inference latency remains nearly constant regardless of the number of retrieved segments, facilitating practical robot deployment.

## 2. Related Work

**Vision-Language-Action (VLA) Models.** The paradigm of robotic control has shifted from task-specific learning toward the development of generalist policies that leverage internet-scale knowledge. Central to this shift is the emergence of VLA models, which integrate large-scale Vision-Language Model (VLM) (Karamcheti et al., 2024; Steiner et al., 2024; Chen et al., 2025) priors with diverse robotic datasets (Walke et al., 2023; Khazatsky et al., 2024; O'Neill et al., 2024). Early studies (Zitkovich et al., 2023; Kim et al., 2025b; Pertsch et al., 2025) rely on action discretization to frame robotic manipulation as a multimodal sequence modeling problem, predicting commands as discrete tokens within a fixed vocabulary. Expanding upon this foundation, more recent approaches (Li et al., 2024; Kim et al., 2025a; Black et al., 2025b;a; Bjorck et al., 2025) have adopted generative action heads—incorporating diffusion or flow-matching mechanisms—to better capture the complex, multi-modal nature of robotic trajectories. Despite these advancements, recent studies (Guruprasad et al., 2024; Fang et al., 2025; Fei et al., 2025; Gu et al., 2025) have exposed a persistent fragility when deployed on novel tasks, often suffering from a severe performance collapse under minor environmental perturbations. This lack of robust generalization underscores the necessity for dynamic, in-context adaptation to mitigate epistemic uncertainty in unseen tasks.

**In-Context Imitation Learning (ICIL).** ICIL represents a viable pathway to achieve test-time adaptation without requiring expensive weight updates. Early efforts in this domain, including one-shot imitation learning, pioneered the use of expert demonstrations as few-shot guidance to steer robotic policies. However, these methods are intrinsically tied to non-RGB modalities (e.g., depth or point clouds) (Vitiello et al., 2023; Biza et al., 2023; Palo & Johns, 2024; Vosylius & Johns, 2025), or lack compatibility with modern VLA models (Duan et al., 2017; Fu et al., 2025; Yoo et al., 2025). This structural rigidity precludes the full utilization of the rich semantic priors inherent in modern VLAs. While subsequent studies (Zhu et al., 2024; Sridhar et al., 2025) have integrated in-context adaptability into more generalist frameworks, they still face a fundamental adaptation bottleneck, failing to effectively leverage expert demonstrations as actionable guidance. This bottleneck hinders real-world deployment and constrains the practical utility of existing ICIL frameworks.

## 3. Preliminaries

### 3.1. Robotic Manipulation

We consider the problem of language-conditioned robotic manipulation, where a policy $\pi_\theta$ aims to map multimodal inputs to a sequence of control signals (e.g., end-effector

poses and gripper states). Formally, at each timestep $t$, the policy receives a visual observation $\mathbf{V}_t$ (e.g., RGB images from static or eye-in-hand cameras), a language instruction $L$, and the robot's proprioceptive state $\mathbf{s}_t$. Following the action chunking paradigm (Zhao et al., 2023) to ensure temporal consistency, the policy predicts an action chunk $\mathbf{A}_t \in \mathbb{R}^{h \times c}$, where $h$ and $c$ denote the action horizon and action dimension, respectively. This mapping is defined as:

$$\mathbf{A}_t = [\mathbf{a}_t, \mathbf{a}_{t+1}, ..., \mathbf{a}_{t+h-1}] = \pi_\theta(\mathbf{V}_t, L, \mathbf{s}_t). \quad (1)$$

The policy is trained via behavioral cloning to minimize the discrepancy between the predicted action chunk and the expert demonstration chunk.

## 3.2. Flow-Matching-Based VLA Architecture

We employ a flow-matching-based Vision-Language-Action (VLA) architecture (Bjorck et al., 2025) that bridges high-level semantic perception with low-level generative control. The model utilizes a Vision-Language Model (VLM) $f_\phi$ to encode the visual observation $\mathbf{V}_t$ and language instruction $L$, producing multimodal features $\mathbf{H}_t = f_\phi(\mathbf{V}_t, L)$.

The action head $g_\psi$ is a Diffusion Transformer (Peebles & Xie, 2023) that refines a noise distribution into the target action chunk. Conditioned on the multimodal features $\mathbf{H}_t$ and the robot's proprioceptive state $\mathbf{s}_t$, the action head regresses the vector field $\mathbf{F}_t = (\mathbf{A}_t - \mathbf{A}_t^{(0)})$ to guide the denoising process:

$$\mathbf{A}_t^{(k+\Delta k)} = \mathbf{A}_t^{(k)} + \Delta k \cdot g_\psi(\mathbf{H}_t, \mathbf{s}_t, \mathbf{A}_t^{(k)}, k), \quad (2)$$

where $k \in [0, 1]$ is the flow-matching timestep and $\mathbf{A}_t^{(k)}$ is the noisy action chunk. This decoupled design ensures precise motor control by grounding the generative process in rich semantic features.

## 4. Method

To overcome the limitations of existing in-context imitation learning (ICIL) approaches, we propose RA-VLA, a retrieval-augmented VLA framework that empowers a policy to perform novel manipulation tasks with negligible computational overhead. This section first outlines the overall architecture of our framework and then details the specific components that enable robust in-context imitation. To rigorously evaluate the policy's training-free adaptability, we maintain a strict separation between the training[1] and evaluation task sets. Further details regarding the framework are provided in Appendix B.

---

[1] In this context, training refers to the process of endowing the VLA policy with in-context adaptability.

### 4.1. Overall Framework

The primary objective of RA-VLA is to facilitate training-free adaptation to novel tasks by leveraging a sparse set of expert demonstrations (e.g., 1–5 demonstrations per task) stored in a buffer $\mathcal{B}$. As illustrated in Figure 2, the framework operates through a systematic pipeline of expert segment retrieval and grounded action generation.

**Context Buffer Construction.** To populate the context buffer $\mathcal{B}$, RA-VLA slices long-horizon expert demonstrations into functional segments $(\mathbf{V}', L', \mathbf{s}', \mathbf{A}')$ by applying a sliding window of length $h$ with a stride of $s$. These segments serve as expert guidance to steer the policy toward novel tasks. To process this guidance without expanding the input sequence length, the framework treats each segment as an independent encoding unit, thereby bypassing the prohibitive computational overhead of full concatenation. This independence, coupled with a frozen VLM backbone $f_\phi$, allows the multimodal features $\mathbf{H}' = f_\phi(\mathbf{V}', L')$ for all segments to be pre-computed and cached offline. Consequently, each segment is stored in the buffer as a key-value pair: the key $\mathbf{z}'$ is used for retrieval, while the corresponding value $(\mathbf{H}', \mathbf{s}', \mathbf{A}')$ provides the in-context guidance.[2]

**Expert Segment Retrieval.** For a given query consisting of a visual observation $\mathbf{V}_t$ and a language instruction $L$, a retrieval encoder $R(\cdot)$ retrieves the $K$ most relevant expert segments $\mathcal{H}_{\text{ret}} = \{\mathbf{H}_{\text{ret}}^{(1)}, ..., \mathbf{H}_{\text{ret}}^{(K)}\}$ from the buffer $\mathcal{B}$. The retrieval encoder $R(\cdot)$ is a lightweight, two-layer Transformer encoder (Vaswani et al., 2017). It directly takes the multimodal input tokens that are fed into the VLM's LLM backbone, which include image embeddings from the vision encoder and text token embeddings. The resulting tokens are average-pooled into embedding $\mathbf{z}$, from which the top-$K$ segments are retrieved based on their cosine similarity to the query.

**Grounded Action Generation.** The action head $g_\psi$ generates control sequences by leveraging the retrieved expert segments $\mathcal{H}_{\text{ret}}$ alongside the current observation $\mathbf{H}_t$:

$$\mathbf{F}_t = g_\psi(\mathbf{H}_t, \mathcal{H}_{\text{ret}}, \mathbf{s}_t, \mathbf{A}_t^{(k)}, k). \quad (3)$$

Instead of attending to the entire set of expert features in every layer, our pairwise grounding strategy concatenates current observation features $\mathbf{H}_t$ with a single expert feature $\mathbf{H}_{\text{ret}}^{(i)}$ for each cross-attention layer. The expert segments are distributed progressively across layers, with the highest layers assigned the most relevant segments for fine-grained action refinement. This design maintains a minimal computational overhead while leveraging the full retrieved context.

---

[2] In practice, the VLM backbone processes visual observations and language instructions, while the robot's proprioceptive state and action sequences are encoded using separate linear encoders.

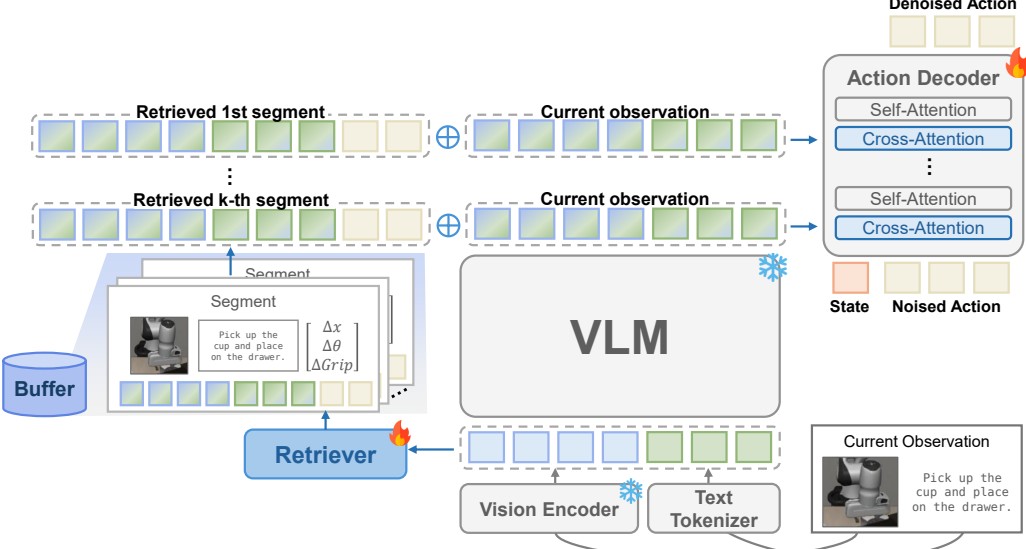

*Figure 2.* **Overview of RA-VLA.** Our framework retrieves task-relevant expert segments to serve as in-context guidance, facilitating action generation for unseen task adaptation. The architecture integrates two key components: a retriever trained to identify behaviorally similar segments (Section 4.2) and a VLA policy finetuned to effectively ground action generation on the retrieved context (Section 4.3).

## 4.2. Learning Action-Aware Retrieval

To ensure the retrieval process captures behavioral relevance rather than mere visual similarity, we propose a *behavioral alignment loss* for the retrieval encoder $R(\cdot)$. The primary objective is to align the embedding space such that segments with similar behavioral patterns are mapped closely together, remaining robust to minor visual variations.

**Behavioral Pairs.** To obtain the behaviorally similar segment pairs, we randomly sample two expert demonstrations of the same task and align their action sequences via Dynamic Time Warping (DTW) (Berndt & Clifford, 1994). This non-linear warping mechanism matches the structural profiles of the trajectories, which inherently accommodates both temporal stretching and spatial translation. Consequently, the aligned segment pairs share behavioral correspondence, regardless of visual appearance. Since such action sequences are unavailable within test-time observations, we train the retrieval encoder to approximate this action-based alignment using solely observation inputs.

**Behavioral Alignment Loss.** Using the alignments derived from DTW, we perform contrastive learning to train the retrieval encoder $R(\cdot)$. We optimize the encoder using a contrastive loss (Chen et al., 2020):

$$\mathcal{L}_{\text{align}} = - \sum_{(i,j)\in\mathcal{P}} \log \frac{\exp(\text{sim}(\mathbf{z}_i, \mathbf{z}_j)/\tau)}{\sum_{k\in\{j\}\cup\mathcal{N}_i} \exp(\text{sim}(\mathbf{z}_i, \mathbf{z}_k)/\tau)}, \tag{4}$$

where $\text{sim}(\cdot, \cdot)$ denotes cosine similarity and $\tau$ is a temperature hyperparameter. Here, $\mathcal{P}$ represents the set of positive pairs $(i, j)$ that are aligned via the DTW optimal path. The set $\mathcal{N}_i$ denotes the negative samples for anchor $i$, which includes all other segments in the training batch (i.e., segments that are not aligned with anchor $i$ through DTW, as well as segments from different tasks). Through this alignment loss, $R(\cdot)$ learns to identify expert segments that provide behavioral guidance for the current observations. This formulation enables the behaviorally aligned retrieval to generalize to unseen tasks, as the retriever encoder captures underlying motion profiles that are not confined to the specific tasks encountered during training.

## 4.3. Learning Context-Grounded Action Generation

In-context imitation learning often faces the challenge of *behavioral inertia*, where the policy relies heavily on its pre-trained priors and disregards the contextual guidance. To encourage the action head $g_\psi$ to ground its predictions in the retrieved expert segments, we introduce a *contextual adherence loss* based on a regression margin mechanism.

**Contextual Adherence Loss.** During each training iteration, we compute two distinct mean squared error (MSE) losses with respect to the target vector field $\mathbf{F}_t$: the relevant context loss $\mathcal{L}_{\text{rel}}$ utilizing segments retrieved via $R(\cdot)$, and the irrelevant context loss $\mathcal{L}_{\text{irrel}}$ utilizing randomly sampled segments:

$$\mathcal{L}_{\text{rel}} = \|\mathbf{F}_t - g_\psi(\mathbf{H}_t, \mathcal{H}_{\text{ret}}, \mathbf{s}_t, \mathbf{A}_t^{(k)}, k)\|^2,$$
$$\mathcal{L}_{\text{irrel}} = \|\mathbf{F}_t - g_\psi(\mathbf{H}_t, \mathcal{H}_{\text{rand}}, \mathbf{s}_t, \mathbf{A}_t^{(k)}, k)\|^2. \tag{5}$$

Using these, the contextual adherence loss is defined as:

$$\mathcal{L}_{\text{adhere}} = \max(0, m - (\mathcal{L}_{\text{irrel}} - \mathcal{L}_{\text{rel}})), \qquad (6)$$

where $m > 0$ denotes a fixed margin. By design, $\mathcal{L}_{\text{adhere}}$ incentivizes the policy to ground its predictions in the expert segments. If the policy disregards the retrieved context and predicts actions solely from current observations, $\mathcal{L}_{\text{rel}}$ and $\mathcal{L}_{\text{irrel}}$ collapse to similar values, violating the margin $m$ and incurring a loss penalty. From an efficiency standpoint, $\mathcal{L}_{\text{adhere}}$ adds a minor training cost by reusing multimodal features, while introducing zero inference overhead.

The overall optimization objective for fine-tuning the policy $\pi_\theta$ is formulated as:

$$\mathcal{L}_{\text{overall}} = \mathcal{L}_{\text{rel}} + \lambda \mathcal{L}_{\text{adhere}}, \qquad (7)$$

where $\lambda$ serves as a weighting hyperparameter that balances the action regression loss $\mathcal{L}_{\text{rel}}$ and the contextual adherence loss $\mathcal{L}_{\text{adhere}}$.

**Remark.**  While conventional flow-matching formulations typically sample the initial noisy action chunk $\mathbf{A}_t^{(0)}$ from a standard normal distribution, we instead initialize $\mathbf{A}_t^{(0)}$ with the empirical mean action chunk of the retrieved expert segments. This informed behavioral prior accelerates training convergence and significantly stabilizes the denoising process on unseen tasks.

## 5. Experiments

### 5.1. Experimental Setups

**Environments.**  We evaluate the in-context adaptability of RA-VLA across the LIBERO benchmark (Liu et al., 2023) and a real-world UR5e robot environment. LIBERO is a comprehensive robot learning benchmark that covers a diverse range of manipulation scenarios. In our experiments, we utilize four task suites—LIBERO-Spatial, LIBERO-Object, LIBERO-Goal, LIBERO-Long—each consisting of 10 distinct tasks with 50 expert demonstrations per task. Our real-world UR5e environment is designed to replicate daily activities, comprising four representative tasks that require varied manipulation skills: (1) **Stack Box**: stacking a small box onto a larger box, (2) **Throw Trash**: throwing a trash into a red bin, (3) **Close Drawer**: closing an open drawer, (4) **Press Pedal**: pressing a pedal to open a trash can. For each task, we collected 30 expert demonstrations using the GELLO teleoperation interface (Wu et al., 2024). To account for environmental variability, each trial involved slight perturbations in the placement of the target and distractor objects, alongside minor variations in the robot's initial proprioceptive states.

**Evaluation Protocol.**  To rigorously evaluate the framework's training-free adaptability to novel tasks, we strictly decouple the training and evaluation tasks across all experimental environments. Within the LIBERO benchmark, we utilize three of the four available task suites to endow in-context capabilities, reserving the remaining suite as an entirely unseen domain for evaluation. For each held-out task, the buffer is populated with three expert demonstrations to serve as in-context guidance, and the performance is measured as the mean success rate over 50 trials. Similarly, in the real-world UR5e environment, we follow a leave-one-out scheme: three of the four tasks are used for training, while the remaining task is held out for evaluation. To enrich the diversity of learned manipulation priors, we augment the three in-domain training tasks with the LIBERO datasets as an auxiliary training source. During evaluation, the buffer is populated with four expert demonstrations for each held-out task to provide the necessary in-context guidance, and the evaluation results are reported as the success rate over 12 trials per task.

**Baselines.**  We compare the performance of RA-VLA against the following baselines:[3]

- **Vanilla VLA** is trained only on the designated training tasks. It measures the VLA's inherent generalization to novel tasks without external guidance.

- **RAEA** (Zhu et al., 2024) conditions the policy by concatenating expert segments directly into the multimodal input prompt, allowing the policy to infer motion from the expert's behavior.

- **RICL** (Sridhar et al., 2025) fuses generative predictions with expert actions via a distance-weighted interpolation. Based on the weighting parameter, we evaluate two variants: **RICL$_G$**, which prioritizes the VLA's generative output, and **RICL$_R$**, which assigns higher weight to the retrieved expert action.

**Implementation Details.**  Across all evaluated frameworks, we utilize GR00t N1.5 (Bjorck et al., 2025) as the underlying VLA backbone to ensure a consistent comparison. For the visual observation, the model processes dual-view RGB images from a third-person camera and a wrist camera, providing both global context and fine-grained local details. Regarding the action generation parameters, we maintain a consistent action horizon of 16 with 4 denoising steps across all experimental configurations. Unless otherwise specified, our default setup augments the policy with a single retrieved expert segment, serving as the primary in-context guidance. The detailed hyperparameters are provided in Appendix E.

---

[3]Detailed description can be found in Appendix D.

*Table 1.* **Novel task adaptation performance on the LIBERO benchmark.** All methods are evaluated on an entirely unseen task suite, utilizing three expert demonstrations per each held-out task as in-context guidance. Baselines are evaluated with both off-the-shelf SigLIP 2 and our SigLIP 2-based action-aware retriever. We report the mean success rate (%) for each task suite, and Average denotes the average success rate across all task suites. **Bold** and underline indicate the best and second-best performances, respectively.

| Retriever | Method | LIBERO-Spatial | LIBERO-Object | LIBERO-Goal | LIBERO-Long | Average |
|---|---|---|---|---|---|---|
| – | Vanilla VLA | 0.000 | 0.066 | 0.002 | 0.000 | 0.0170 |
| Off-the-shelf (SigLIP 2) | RAEA | 0.060 | 0.106 | 0.008 | 0.006 | 0.0450 |
| | $RICL_G$ | 0.092 | 0.156 | 0.106 | 0.000 | 0.0885 |
| | $RICL_R$ | 0.048 | 0.052 | 0.086 | 0.000 | 0.0465 |
| Action-aware (Ours) | RAEA | 0.164 | 0.182 | 0.104 | 0.044 | 0.1235 |
| | $RICL_G$ | 0.156 | 0.198 | 0.116 | 0.048 | 0.1295 |
| | $RICL_R$ | 0.210 | 0.326 | 0.206 | 0.092 | 0.2085 |
| | **RA-VLA** | **0.320** | **0.556** | **0.532** | **0.130** | **0.3845** |

*Table 2.* **Novel task adaptation performance on the real-world UR5e environment.** All methods are evaluated on an entirely unseen task suite, utilizing four expert demonstrations per held-out task as in-context guidance. Baselines are evaluated with our SigLIP 2-based action-aware retriever. We report both sub-goal and overall success rates for each task.

| | Stack Box | | Throw Trash | | Close Drawer | | Press Pedal | | |
|---|---|---|---|---|---|---|---|---|---|
| Method | *Pick Box* | Success | *Pick Trash* | Success | *Touch Drawer* | Success | *Touch Pedal* | Success | Average |
| Vanilla VLA | 0.000 | 0.000 | 0.083 | 0.000 | 0.500 | 0.333 | 0.000 | 0.000 | 0.0833 |
| RAEA | 0.000 | 0.000 | 0.250 | 0.000 | 0.667 | 0.500 | 0.000 | 0.000 | 0.1250 |
| $RICL_R$ | 0.583 | 0.250 | 0.833 | 0.250 | 0.667 | 0.583 | **0.667** | 0.333 | 0.3542 |
| **RA-VLA** | **0.750** | **0.417** | **0.917** | **0.750** | **0.750** | **0.667** | 0.583 | **0.417** | **0.5625** |

## 5.2. Novel Task Adaptation

**Simulation Evaluation.** As summarized in Table 1, vanilla VLA models exhibit near-zero success rates on novel tasks, confirming their lack of inherent adaptability to unseen scenarios. While existing ICIL methods utilizing off-the-shelf retriever such as SigLIP 2 (Tschannen et al., 2025) show marginal gains, their performance remains constrained by the frequent retrieval of behaviorally irrelevant segments that act as functional distractors. In contrast, integrating our proposed action-aware retrieval encoder consistently enhances the performance of all evaluated baselines. This consistent improvement across baselines demonstrates that our retrieval mechanism successfully filters out functional distractors in favor of behaviorally relevant context. Notably, RA-VLA, which couples this precise retrieval with enhanced contextual adherence to the expert segments, achieves the highest success rates across all evaluated task suites, improving the success rate from 20.9% to 38.5%. This result validates the synergy between behavioral intent alignment and grounded action execution, demonstrating that retrieved expert knowledge is faithfully integrated into the robot's control loop to guide unseen tasks.

**Real-World Evaluation.** We further validate our findings on a real-robot setup using RAEA, $RICL_R$, and RA-VLA equipped with our action-aware retriever. As shown in Table 2, RA-VLA marks a significant leap in performance, boosting the success rate to 56.3% from the current state-of-

the-art of 35.4%. To qualitatively analyze the performance gains, we examine policy rollouts in Figure 3, focusing on the behavioral characteristics that distinguish RA-VLA from the baselines. For the unseen task of *'putting the small box on the large box'*, RAEA incorrectly picks up the trash instead of the small box. This demonstrates that RAEA tends to favor behaviors prevalent in its training distribution over the provided expert guidance. $RICL_R$, which simply blends retrieved action chunk with generative predictions, manages to lift the box but fails to accurately transport it to the target location. This suggests that such a naive ensemble mechanism yields imprecise and jittery control, preventing the policy from achieving the fine-grained manipulation required for successful task completion. In contrast to these baselines, RA-VLA achieves a robust synergy between expert adherence and adaptive precision, ensuring successful task completion even under significant distributional shifts.

## 5.3. Contextual Sensitivity Analysis

To quantify the degree to which a model relies on the provided expert segments, we measure Relative Contextual Sensitivity $\mathcal{S}_{ctx}$ across the evaluated frameworks. This metric is defined as the relative change in the model's action predictions when the retrieved expert context ($\mathcal{D}_{ret}$) is replaced with a randomly sampled context ($\mathcal{D}_{rand}$):

$$\mathcal{S}_{ctx} = \frac{\|\pi_\theta(\mathbf{V}_t, L, \mathcal{D}_{ret}) - \pi_\theta(\mathbf{V}_t, L, \mathcal{D}_{rand})\|}{\|\pi_\theta(\mathbf{V}_t, L, \mathcal{D}_{ret})\|}. \quad (8)$$

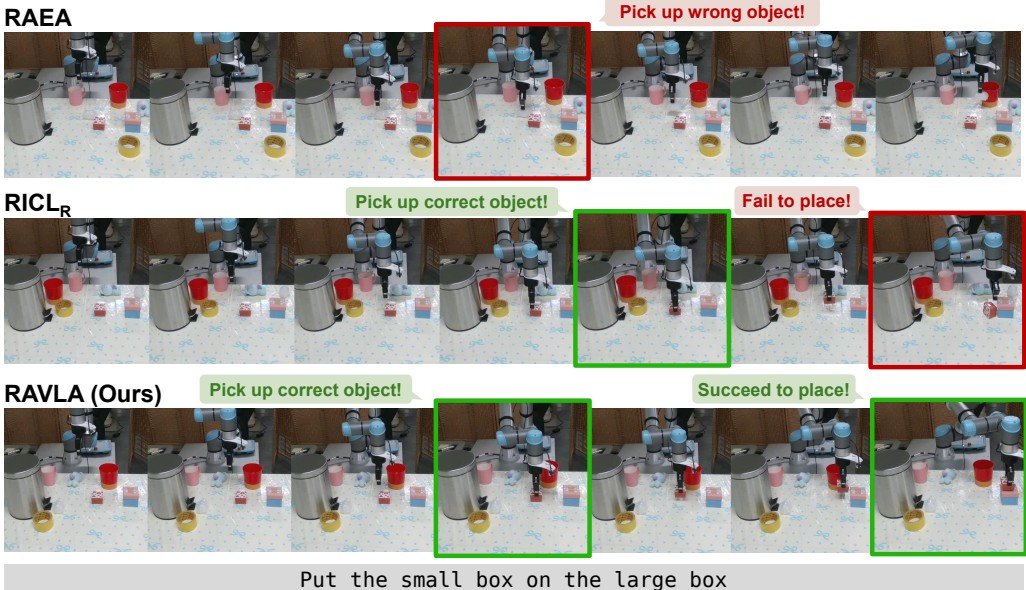

*Figure 3.* **Qualitative comparison of policy rollouts on the real-robot *Stack Box* task.** While all frameworks utilize our action-aware retriever to obtain expert context, only RA-VLA precisely executes unseen instructions by effectively leveraging this guidance. In contrast, baselines fail by reverting to pre-trained priors or lacking control precision. For further qualitative examples, see Figures 6–9.

*Table 3.* **Contextual sensitivity.** The Relative Contextual Sensitivity $\mathcal{S}_{\text{ctx}}$ is averaged over 1,000 samples. We observe a positive trend where $\mathcal{S}_{\text{ctx}}$ correlates with increased success rates.

| Method | $\mathcal{S}_{\text{ctx}}$ | LIBERO-Goal |
|---|---|---|
| RAFT | 0.0247 | 0.104 |
| RICL$_R$ | 0.0788 | 0.206 |
| RA-VLA *w/o* $\mathcal{L}_{\text{adhere}}$ | 0.0353 | 0.098 |
| RA-VLA | 0.3639 | 0.532 |

A higher $\mathcal{S}_{\text{ctx}}$ indicates that the model's *generative prediction* is more tightly coupled with the expert guidance provided in the context.

As shown in Table 3, existing ICIL baselines exhibit significantly lower Relative Contextual Sensitivity than RA-VLA, indicating a lack of contextual grounding during action generation. This contextual neglect stems from the standard behavior cloning objective; since current observations often suffice to minimize the regression loss within the training distribution, the model lacks the incentive to learn query-context interactions. Furthermore, we conduct an ablation study to analyze the impact of our proposed contextual adherence loss $\mathcal{L}_{\text{adhere}}$. When this loss is removed, RA-VLA shows a marked decrease in $\mathcal{S}_{\text{ctx}}$, accompanied by a corresponding drop in success rate. These results demonstrate that our adherence loss is essential for translating the retrieved expert knowledge into context-aware action execution. Further qualitative analysis supporting this observation is provided in Figure 10.

## 5.4. Retriever Analysis

We conduct a qualitative analysis of the retrieval process by visualizing expert segments identified by our action-aware retrieval encoder and off-the-shelf foundation models, such as SigLIP 2 (Tschannen et al., 2025) and Eagle 2 (Chen et al., 2025). Specifically, given a pair of distinct expert trajectories, we examine the segment-wise alignment between them by retrieving the most relevant segment from one trajectory for each segment in the other.

As shown in Figure 4, off-the-shelf models often retrieve behaviorally irrelevant distractors due to the semantic gap between general-purpose visual representations and the functional dynamics for robotic manipulation. In contrast, our action-aware retriever establishes robust correspondence by leveraging representations that encapsulate the underlying action dynamics learned via behavioral alignment loss. By replacing the baseline retriever with our action-aware retriever, the success rate of RA-VLA surges from 10.2% to 53.2% on LIBERO-Goal, demonstrating the critical role of behavioral consistency for in-context adaptation.

## 5.5. Inference Scalability Analysis

Inference latency is a key consideration for high-frequency robotic control, as delays in the control loop can impact the stability of task execution. To evaluate the inference scalability of our framework, we compare the latency of RA-VLA against ICIL baselines as a function of the number of augmented expert segments $K$.

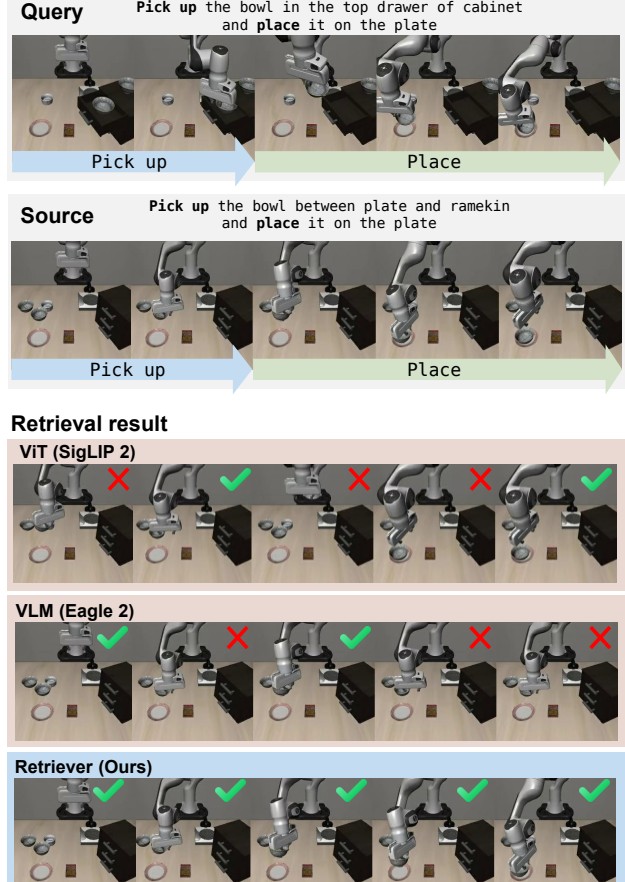

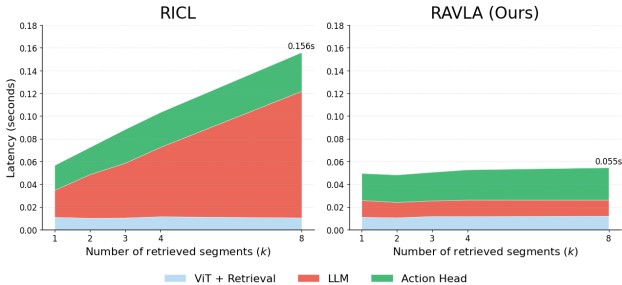

*Figure 5.* **Inference scalability.** The latency is averaged over 1,000 inference runs for the generation of a single action chunk. We use 4 denoising steps for the action head. Unlike existing ICIL approaches, RA-VLA enables the seamless integration of multiple expert segments. RAEA shows a similar trend to RICL, as illustrated in Figure 11.

*Table 4.* **Impact of buffer size $N$ and retrieval size $K$.** We report the average success rate (%) on the LIBERO-Goal.

|   | 1 | 2 | 3 | 4 |
|---|---|---|---|---|
| N | 0.482 | 0.518 | 0.532 | 0.552 |
| K | 0.532 | 0.540 | 0.548 | 0.546 |

*Figure 4.* **Visualization of segment-level trajectory alignment.** For a given query segment, the retrieval model searches for the most relevant segment within the source trajectory. For instance, for the first segment of the query trajectory, the ViT-based retriever identifies the first image in the retrieval results as the most relevant match. However, while the query segment depicts the initial pick-up phase, the retrieved segment corresponds to a later transport phase, illustrating a failure in capturing behavioral relevance.

## 5.6. Analysis of Buffer and Retrieval Size

We further investigate the sensitivity of RA-VLA to the number of expert demonstrations in the buffer $N$ and the number of retrieved expert segments for in-context guidance $K$. As illustrated in Table 4, the success rate increases consistently with the buffer size $N$; a larger demonstration buffer provides a richer search space, enabling the retriever to search for the segments that closely align with the behavioral nuances of the current observation. Regarding the number of retrieved segments $K$, we observe a marginal performance gain as $K$ increases. While a small number of expert segments offer sufficient functional cues in our current setup, the inherent capability to scale $K$ remains highly promising for complex or ambiguous scenarios.

## 6. Conclusion

In this work, we investigated the primary bottlenecks of existing in-context imitation learning (ICIL) frameworks, specifically the systematic inability to transfer the collected expert knowledge into the novel task execution. To overcome this limitation, we introduced RA-VLA, a retrieval-augmented framework that integrates a behavior-aligned retrieval into an efficient, grounded execution pipeline. Extensive experiments across simulation and real-world environments demonstrate that RA-VLA successfully adapts to unseen tasks while bypassing the latency penalty inherent in existing ICIL approaches.

As illustrated in Figure 5, while RICL exhibits a steep increase in latency as more segments are integrated, RA-VLA maintains a near-constant inference time with only negligible overhead. A component-wise breakdown reveals that the primary bottleneck in existing ICIL methods stems from directly appending expert segments to the multimodal input prompt. This strategy leads to a substantial expansion of the context window, causing the computational overhead of the LLM backbone to scale unfavorably with $K$. In contrast, RA-VLA circumvents this bottleneck by treating each retrieved segment as an independent encoding unit. This strategy decouples the inference latency from the scale of retrieved context, successfully mitigating the latency constraints that hinders the practical deployment of in-context policy adaptation.

## Acknowledgements

This work was supported by Samsung Research Funding & Incubation Center of Samsung Electronics under Project Number SRFC-IT2402-05.

## Impact Statement

Our work contributes to the development of more reliable and safe autonomous systems. A primary bottleneck in the real-world deployment of robotic agents is their limited generalization to out-of-distribution scenarios, which can lead to unpredictable or unsafe behaviors in critical environments. By enabling training-free in-context adaptation, RA-VLA provides a mechanism for human operators to quickly adjust a robot's policy through corrective demonstrations.

While RA-VLA enhances adaptability, its reliance on a demonstration buffer introduces a potential vulnerability to adversarial data injection. If a malicious actor were to introduce deceptive or harmful expert demonstrations into the retrieval buffer, the framework might integrate this guidance into its execution. To mitigate these risks, future deployments should implement robust verification protocols to ensure the integrity and quality of the demonstration sources before they are used for real-world robotic control.

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

# A. Extended Related Work

**In-Context Adherence.**    In natural language processing, extensive research has focused on ensuring that language models faithfully utilize information embedded within the provided context. For instance, contrastive decoding (Shi et al., 2024; Zhao et al., 2024) amplifies inherent contextual adherence by leveraging the discrepancy between model predictions with and without context. However, these methods assume the base model already exhibits baseline sensitivity to the context and require multiple forward passes, rendering them ill-suited for large-scale VLAs. On the other hand, establishing context-mandatory training scenarios (An et al., 2024) is highly difficult in in-distribution robotic manipulation. Because a standard observation typically encapsulates all necessary cues for action generation, it is practically unfeasible to design a training setup where the policy is forced to rely on the retrieved context to accurately predict actions. While assigning fixed relative weights of losses with and without context (Zhang et al., 2025) represents the closest parallel to our work, such weighting struggles to provide a sustained optimization signal for context differentiation throughout the entire training process.

# B. Additional Details on RA-VLA

**Demonstration Slicing.**    An expert demonstration $[(\mathbf{V}_1, L, \mathbf{s}_1, \mathbf{a}_1), (\mathbf{V}_2, L, \mathbf{s}_2, \mathbf{a}_2), ...]$ is sliced into functional segments $(\mathbf{V}_t, L, \mathbf{s}_t, \mathbf{A}_t)$ at timesteps $t \in \{1, 1+s, 1+2s, ...\}$, where $\mathbf{A}_t$ denotes the action chunk $[\mathbf{a}_t, ..., \mathbf{a}_{t+h-1}]$.

**Retrieval Efficiency.**    To illustrate the computational efficiency of our retrieval system, consider a typical expert demonstration consisting of 128 action steps. This demonstration is sliced into 15 segments, each stored in a GPU-managed buffer with a 1KB retrieval key vector, a 2MB pre-computed multimodal feature tensor, and a 1KB action chunk tensor. Such memory requirements are practically insignificant given the scale of modern GPU memory. In our current implementation, retrieval is performed via a low-latency matrix multiplication between the query $(1 \times 512)$ and the stored keys $(N \times 512)$, followed by a Top-$K$ operation. Notably, our benchmarks on an NVIDIA A100 GPU show a latency of just 0.18ms (0.36% of the total inference latency) for $N = 10^7$, which ensures that the retrieval stage does not introduce a latency bottleneck as the buffer scales. This design can be further scaled, as techniques like ANN indexing can be seamlessly integrated to handle an even larger volume of demonstrations.

**Pairwise Grounding.**    We manually designate the assignment of retrieved segments across the eight cross-attention layers to ensure a gradual and balanced distribution. For instance, when $K = 2$ segments are retrieved, the first four cross-attention layers are assigned the second most relevant segment, while the remaining four layers receive the most relevant segment, yielding the assignment pattern [2, 2, 2, 2, 1, 1, 1, 1]. The segment assignment patterns for other values of $K$ are provided in Table 5. While our current implementation covers $K \leq 8$, this strategy can be extended by dynamically adjusting assignments according to the denoising steps of flow matching.

*Table 5.* **Segment assignment patterns across different values of $K$.**

| K | Segment Assignment |
|---|---|
| 1 | [1, 1, 1, 1, 1, 1, 1, 1] |
| 2 | [2, 2, 2, 2, 1, 1, 1, 1] |
| 3 | [3, 3, 2, 2, 2, 1, 1, 1] |
| 4 | [4, 4, 3, 3, 2, 2, 1, 1] |
| 5 | [5, 4, 3, 3, 2, 2, 1, 1] |
| 6 | [6, 5, 4, 3, 2, 2, 1, 1] |
| 7 | [7, 6, 5, 4, 3, 2, 1, 1] |
| 8 | [8, 7, 6, 5, 4, 3, 2, 1] |

**Behavioral Pairs.**    Given two expert demonstrations $[(\mathbf{V}_1, L, \mathbf{s}_1, \mathbf{a}_1), ...]$ and $[(\mathbf{V}'_1, L', \mathbf{s}'_1, \mathbf{a}'_1), ...]$, we first apply the DTW algorithm to their respective action sequences $[\mathbf{a}_1, \mathbf{a}_2, ...]$ and $[\mathbf{a}'_1, \mathbf{a}'_2, ...]$. This optimization determines which action pairs $(\mathbf{a}_i, \mathbf{a}'_j)$ should be matched to achieve the best behavioral alignment, which yields a warping path consisting of index pairs $(i, j)$ that minimizes the $L_2$ distance-based cumulative matching cost. Consequently, the aligned segment pairs $(\mathbf{V}_i, L, \mathbf{s}_i, \mathbf{A}_i)$ and $(\mathbf{V}'_j, L', \mathbf{s}'_j, \mathbf{A}'_j)$ are designated as positive pairs, whereas unaligned pairs are treated as negatives.

**Contextual Adherence Loss.** Regarding the formulation of $\mathcal{L}_{\text{adhere}}$, we initially explored variants of ratio-based margin loss that constrain the ratio $\mathcal{L}_{\text{irrel}}/\mathcal{L}_{\text{rel}}$. However, as the policy converges and the absolute scale of the MSE diminishes toward zero, ratio-based losses tend to incur numerical instability and a diminishing gap effect, thereby failing to provide a consistent optimization signal. Consequently, we adopt the difference-based margin loss, which provides a robust and stable learning signal throughout the entire training process.

## C. Real-World Experimental Setup

The real-world robot platform consists of a UR5e robotic arm (Universal Robots) equipped with a Robotiq 2F-85 adaptive gripper. For visual observation, we utilize two Intel RealSense D435 cameras: one positioned as a static third-person camera to provide a global view of the workspace, and the other mounted on the wrist to capture localized, egocentric observations.

Expert demonstrations are collected using the GELLO teleoperation interface (Wu et al., 2024), which allows for intuitive and high-fidelity control of the robotic arm. We synchronize data acquisition and command execution at 10 Hz, using the `ur_rtde` (Lindvig et al., 2025) library to interface with the UR5e robot.

Both the training and inference of the VLA model are conducted using a single NVIDIA A100 GPU. The model is served on an A100 node, and the robot controller communicates with this server to receive predicted action signals.

## D. Additional Details on ICIL Baselines

**Retrieval-Augmented Embodied Agents (RAEA)** (Zhu et al., 2024) incorporates expert demontration segments $\mathcal{D}_{\text{ret}}$ by concatenating them with the current query $(\mathbf{V}_t, L)$ into a unified multimodal prompt. The VLM $f_\phi$ then processes this concatenated sequence to extract multimodal features $\mathbf{H}_t$:

$$\mathbf{H}_t = f_\phi(\mathbf{V}_t, L, \mathcal{D}_{\text{ret}}). \tag{9}$$

Conditioned on these features, the action head $g_\psi$ predicts a vector field $\mathbf{F}_t$, which is subsequently used to generate the action chunk $\mathbf{A}_t$.[4]

**Re-training for In-Context Learning (RICL)** (Sridhar et al., 2025), similar to RAEA, integrates ICIL into the VLA framework by employing a supervised fine-tuning strategy that conditions the policy on expert demonstration segments $\mathcal{D}_{\text{ret}}$. A key distinction of RICL is the action interpolation layer, which fuses expert action chunk $\mathbf{A}'$ with the policy's generative output. The final action chunk $\mathbf{A}_t$ is formulated as a distance-weighted ensemble:[5]

$$\mathbf{A}_t = w(d) \cdot \mathbf{A}' + (1 - w(d)) \cdot \pi_\theta(\mathbf{V}_t, L, \mathcal{D}_{\text{ret}}), \tag{10}$$

where $d = \|R(\mathbf{V}_t) - R(\mathbf{V}')\|_2$ represents the $\ell_2$ distance between the query and the nearest neighbor in the retrieval space. The weighting function $w(d) = e^{-\lambda_w d}$ determines the balance between direct imitation and generative prediction.

## E. Additional Implementation Details

For the ICIL baselines, the weighting parameter $\lambda_w$ is set to 1.0 for $\text{RICL}_{\text{G}}$ and 0.2 for $\text{RICL}_{\text{R}}$. In RA-VLA, we set the temperature $\tau$ to 0.01, the stride $s$ to 4, the margin $m$ to 0.01, and the loss weighting factor $\lambda$ to 0.1.[6] The retrieval encoder $R(\cdot)$ is configured with a hidden size of 512, 8 attention heads, an intermediate size of 1024, and 2 hidden layers.

---

[4]The architecture described here is our VLA-adapted implementation of RAEA, which extends its original principles to high-capacity VLA models.

[5]For adaptation to flow matching, we rearrange Equation 10 with respect to the output of the policy $\pi_\theta$ to define the regression loss.

[6]Although not applied to our main experiments, reducing the update magnitude of the noisy action chunk during the denoising steps showed stable performance on the LIBERO benchmark.

## F. Limitations

- Since the buffer serves as the sole source of guidance to unseen tasks, the quality and diversity of the expert demonstrations inherently dictates the in-context adaptability of the policy. Given that the specific demonstrations stored in the buffer notably impact performance, all experiments are conducted using the exact same buffer to ensure a fair comparison.

- The absolute success rate remains low relative to the in-distribution performance, and the performance exhibits high sensitivity to hyperparameters due to the out-of-distribution nature of the evaluation.

- Our findings warrant further validation across a broader range of experimental setups, such as large-scale training, non-flow-matching architectures, and complex manipulation tasks. For instance, as the training scale increases, an independent encoding strategy may constrain performance.

## G. Future Directions

Looking ahead, several promising avenues remain for further exploration. Extending our grounding mechanism to support cross-embodiment adaptation will substantially broaden the applicability of our framework. Furthermore, incorporating more diverse forms of expert guidance, such as unstructured human videos, represents a vital step toward achieving general-purpose robotic intelligence.

## H. Additional Experimental Results

We provide additional policy rollouts across diverse experimental configurations (Figures 6–9), policy rollouts with and without the contextual adherence loss $\mathcal{L}_{\text{adhere}}$ (Figure 10), and the full inference scalability results (Figure 11).

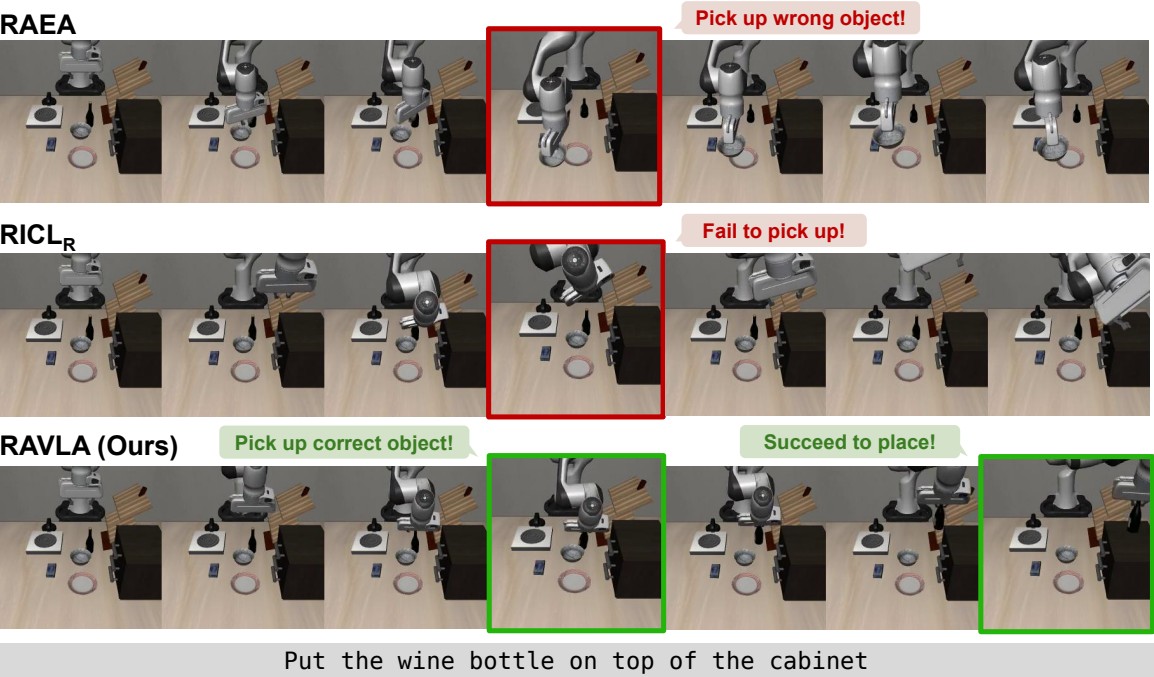

*Figure 6.* **Qualitative comparison of policy rollouts for the *'put the wine bottle on top of the cabinet'* task in LIBERO-Goal.** While all frameworks utilize our action-aware retriever to obtain expert context, only RA-VLA precisely executes unseen instructions by effectively leveraging this guidance. In contrast, baselines fail by reverting to pre-trained priors or lacking control precision.

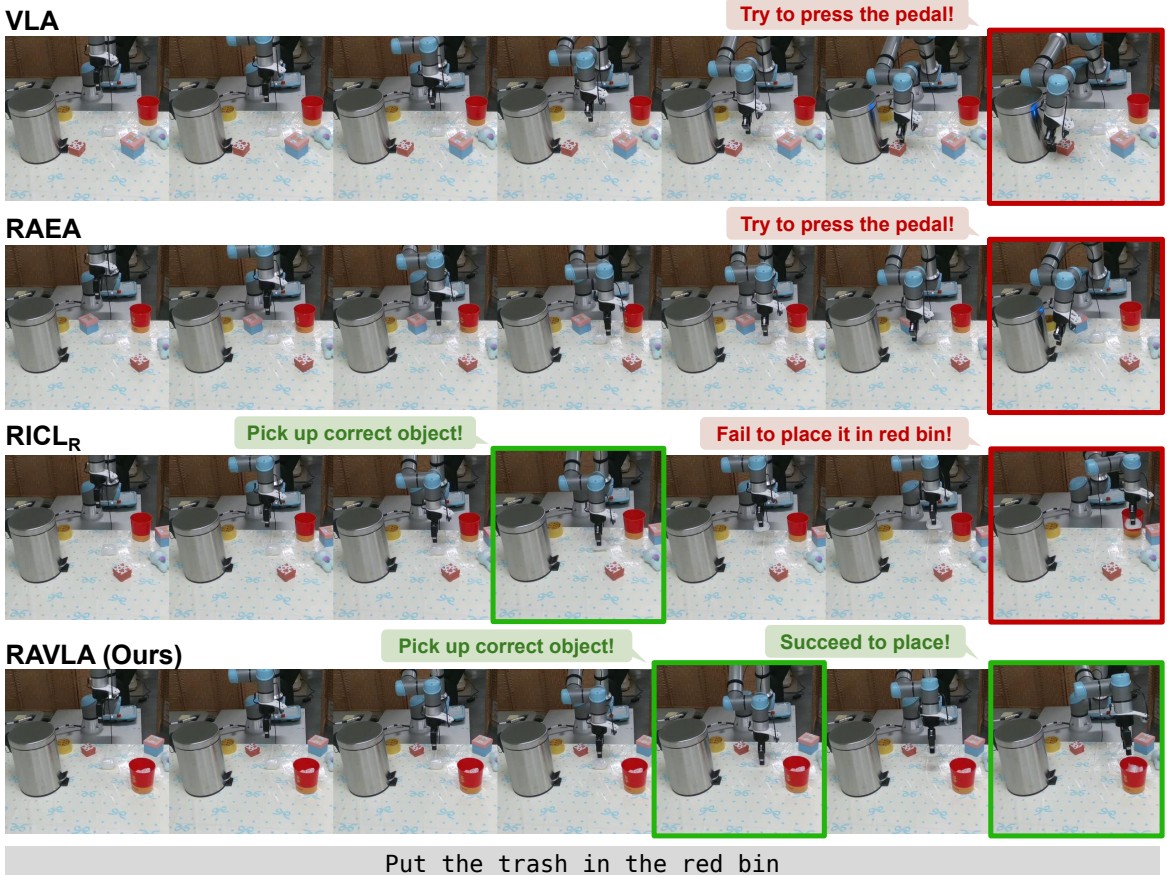

*Figure 7.* **Qualitative comparison of policy rollouts on the real-robot *Throw Trash* task.** While all frameworks utilize our action-aware retriever to obtain expert context, only RA-VLA precisely executes unseen instructions by effectively leveraging this guidance. In contrast, baselines fail by reverting to pre-trained priors or lacking control precision.

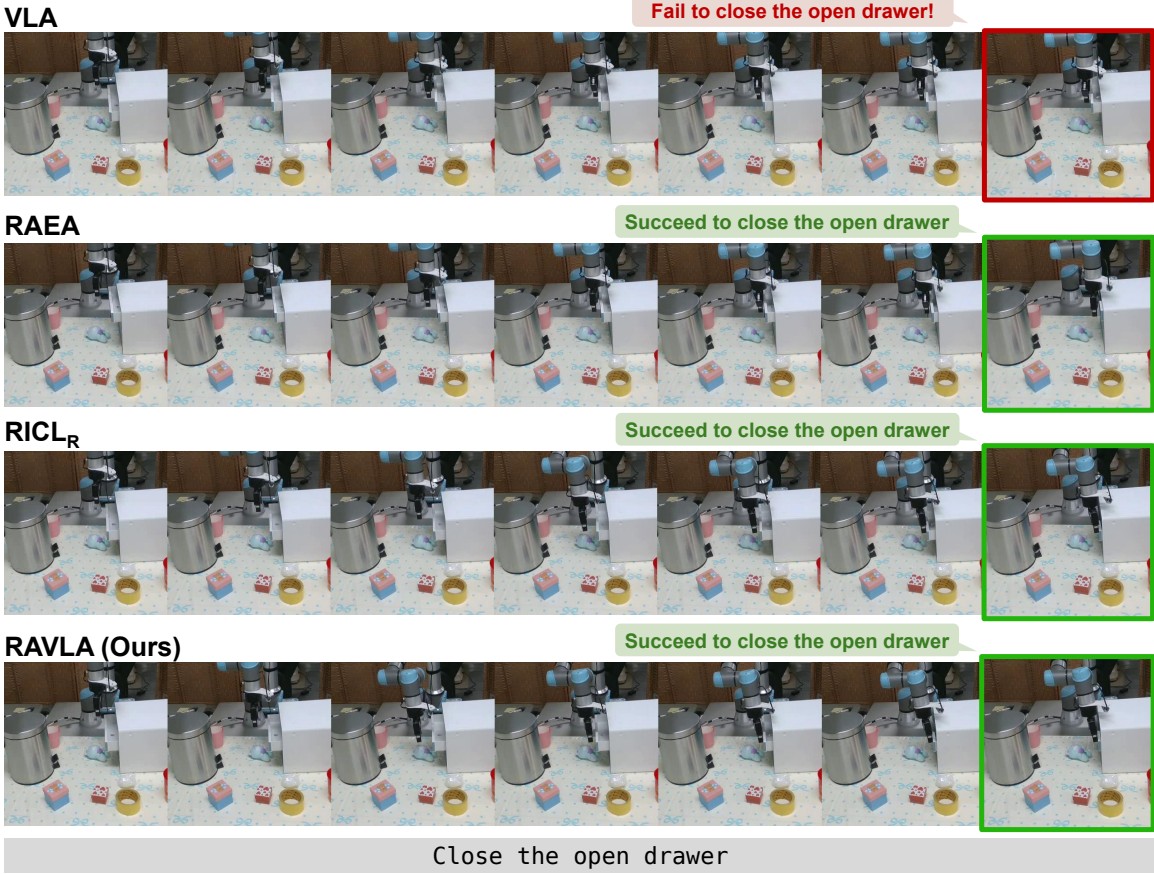

*Figure 8.* **Qualitative comparison of policy rollouts on the real-robot *Close Drawer* task.** For this specific task, the vanilla VLA exhibits a notable degree of adherence to the given instruction, suggesting that its pre-training distribution likely encompasses demonstrations relevant to drawer-closing maneuvers.

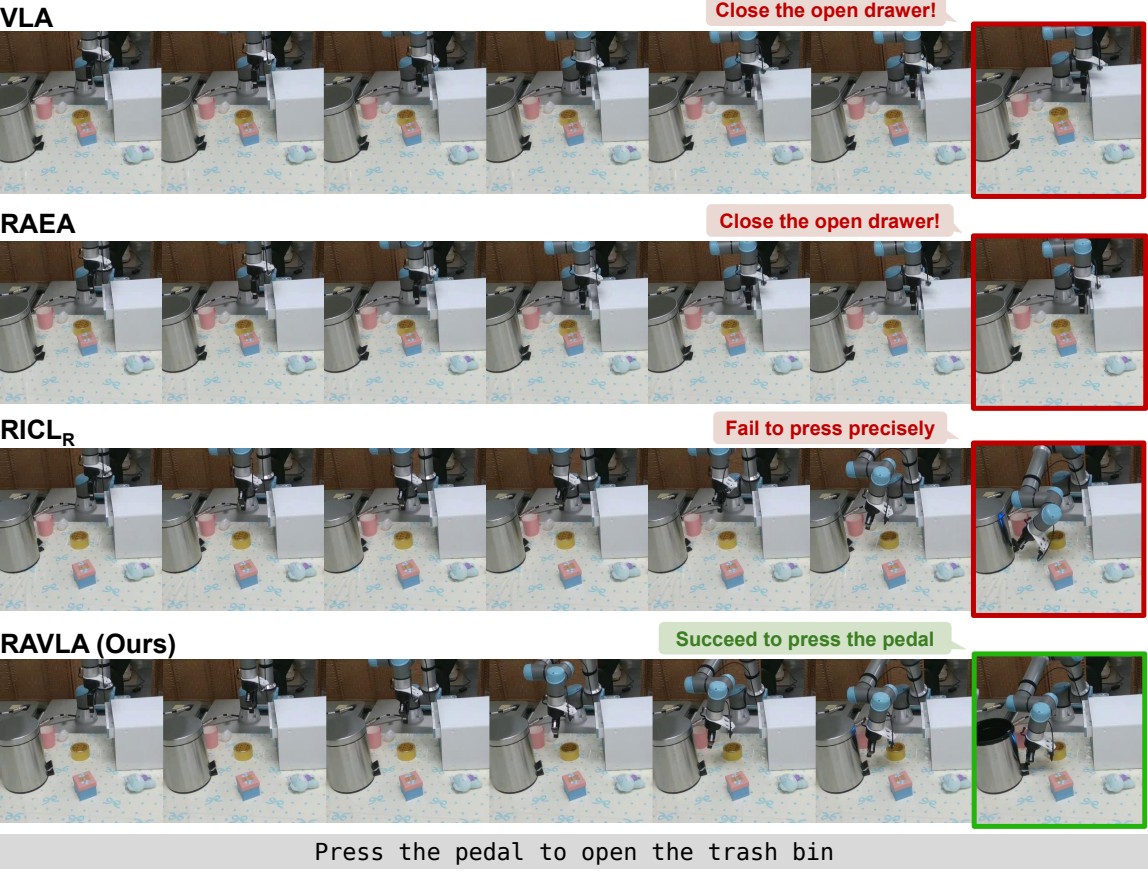

*Figure 9.* **Qualitative comparison of policy rollouts on the real-robot *Press Pedal* task.** While all frameworks utilize our action-aware retriever to obtain expert context, only RA-VLA precisely executes unseen instructions by effectively leveraging this guidance. In contrast, baselines fail by reverting to pre-trained priors or lacking control precision.

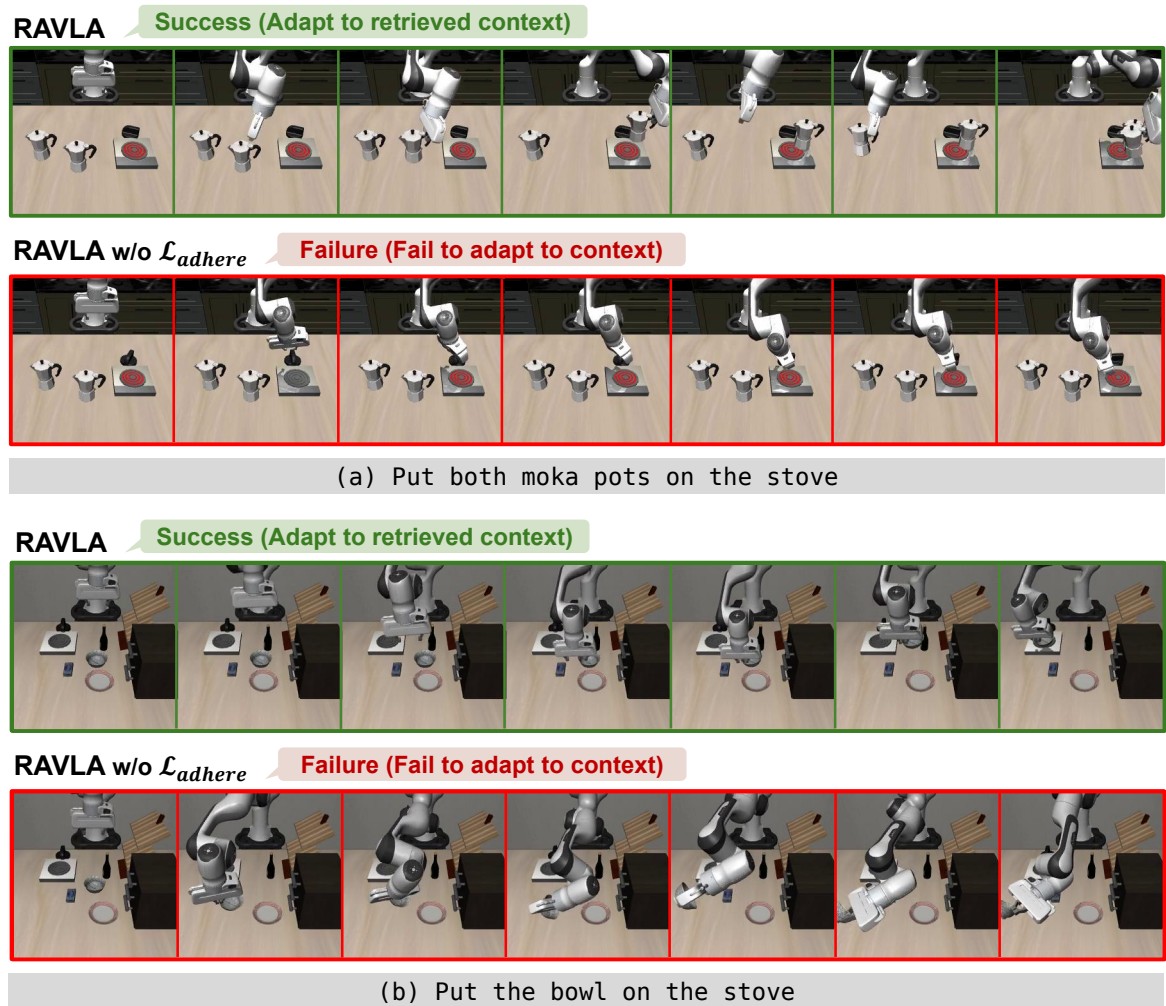

Figure 10. **Qualitative comparison of policy rollouts with and without contextual adherence loss $\mathcal{L}_{\text{adhere}}$.** In the absence of the contextual adherence loss, the policy fails to align with the expert guidance and instead (a) reverts to its pre-trained priors or (b) produce erratic actions.

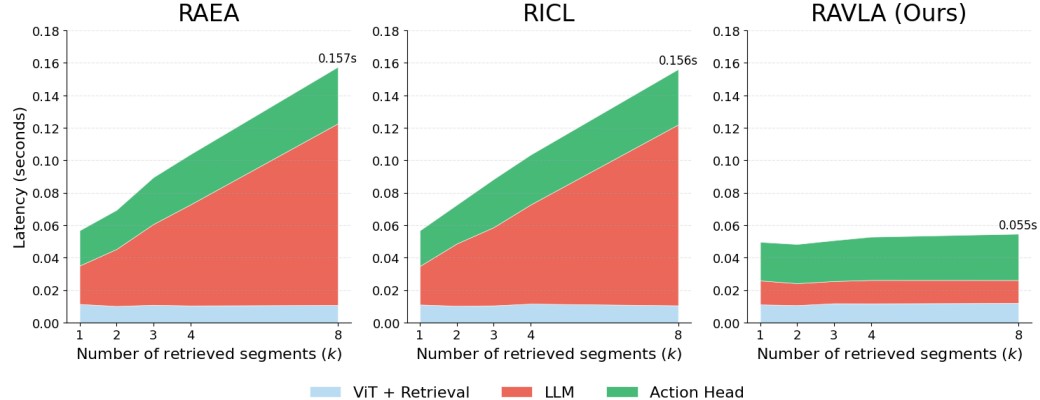

Figure 11. **Inference scalability.** The latency is averaged over 1,000 inference runs for the generation of a single action chunk. We use 4 denoising steps for the action head. Unlike existing ICIL approaches, RA-VLA enables the seamless integration of multiple expert segments.

