# OpenReview forum: "RA-VLA: Retrieval-Augmented VLA for Test-Time Adaptation"
_ICML.cc/2026/Conference — ICML 2026 regular_

### Official Review · Reviewer_WFs2 · 2026-02-25

**Soundness:** 3
**Presentation:** 3
**Significance:** 2
**Originality:** 3
**Overall Recommendation:** 4
**Confidence:** 4

**Summary:**

This paper proposes RA-VLA, a VLA fine-tuning method that aims to solve a major bottleneck in in-context imitation learning: lack of adherence to behaviors presented in expert demonstrations, causing the policy to fall back on what was learned during pretraining. RA-VLA trains a retriever using contrastive loss to differentiate between expert demonstrations by their actual behaviors. During VLA training, this retriever pulls in relevant in-context examples from a history buffer, and a margin loss is used to prioritize relevant in-context examples and enforce a difference in behavior between relevant and irrelevant examples.

**Compliance With Llm Reviewing Policy:**

Affirmed.

**Final Justification:**

My concerns were fully addressed in the rebuttal. I raised my score from 3 (weak reject) to 4 (weak accept).

**Key Questions For Authors:**

- How are trajectories stored in the buffer? What is the exact representation?
- The paper claims latency remains "nearly constant" regardless of the number of retrieved segments. Could the authors clarify the scaling behavior? If K exceeds the number of cross-attention layers, multiple features must be assigned per layer, increasing the cross-attention sequence length. Even below that threshold, each additional segment increases one layer's input. Is the claim of constant latency just an artifact of the action head being cheap relative to the LLM backbone?
- Could the authors provide more detail on how DTW was applied to the action sequences? What distance metric was used between action segments, and how were the alignment paths converted into positive pairs for the contrastive loss?
- The paper frames behavioral inertia (reverting to pre-trained priors) as purely negative, but in many deployment scenarios this could be a safety feature. Could the authors discuss when strong adherence to retrieved context could be harmful, e.g., when the retrieved demonstrations are noisy or adversarial?

If the authors can resolve these questions during the rebuttal period, I will absolutely increase my score.

**Limitations:**

- The paper lacks a dedicated limitations section. Several important limitations go undiscussed, such as the ceiling on K imposed by the number of cross-attention layers, the still-low absolute success rates, and the small number of real-world evaluation trials (12 per task).

**Strengths And Weaknesses:**

## Strengths

- The evaluation setup is clean: two different platforms (simulation and real-world), with a leave-one-out evaluation protocol.
- The overall loss formulation is clean and well-motivated.
- The inference scalability analysis clearly demonstrates the latency advantage of treating retrieved segments as independent encoding units rather than concatenating them into the prompt.



## Weaknesses

- Figure 1(c) does not specify the baselines used in the latency comparison. This should be fixed.
- The related work section could use better treatment. Related work is a chance to show where your approach differs from prior work. Works that involve improving adherence to in-context examples outside of the robotic setting should be briefly discussed since they are highly relevant.
- The pairwise grounding strategy (Section 4.1) is extremely vague. The authors claim that deeper layers focus on the most relevant features, but this seems like an unjustified assumption. Why not include the most relevant features in earlier layers so they are considered in every subsequent layer? How is the relevance ordering determined? This is a core design choice that enables the paper's efficiency claims — if it turns out you need all features at every layer for good performance, the entire constant-latency story collapses. No ablation is provided on the layer assignment strategy (e.g., reversed ordering, all layers getting all features, random assignment).
- The contrastive alignment for the retriever is well-motivated, but the DTW-based behavioral alignment could use more detail on exactly how it was applied.
- Typo on line 194: "togather" → "together."

---

> ### Author Rebuttal · Authors · 2026-03-31
>
> We would like to express our sincere gratitude for your time and effort in evaluating our work. We hope the following responses satisfactorily address your concerns.
>
> &nbsp;
> ### W1. Baseline Specification
> We apologize for the confusion and will explicitly label the baseline in Figure 1(c) as RICL in the revised manuscript.
>
> &nbsp;
> ### W2. Literature Review on In-Context Adherence
> We will incorporate a comprehensive literature review regarding the in-context adherence. Brief summary: (1) Unlike NLP studies [1-2] that amplify inherent contextual adherence at inference, VLA models lack this baseline sensitivity, rendering such method inapplicable. (2) Constructing context-mandatory training scenarios [3] is less effective for in-distribution manipulation tasks, where the current observation already contains all requisite information, leading models to bypass the provided context. (3) While [4] employs a loss-weighting scheme, the optimization pressure to enhance context-sensitivity diminishes as the training loss minimizes. In contrast, our proposed loss introduces a persistent margin that maintains a robust supervisory signal even as training approaches convergence.
> > [1] Trusting Your Evidence: Hallucinate Less with Context-aware Decoding. NAACL 2024.
> >
> > [2] Enhancing Contextual Understanding in Large Language Models through Contrastive Decoding. NAACL 2024.
> >
> > [3] Make Your LLM Fully Utilize the Context. NeurIPS 2024.
> >
> > [4] More is not always better? Enhancing Many-Shot In-Context Learning with Differentiated and Reweighting Objectives. ACL 2025.
>
> &nbsp;
> ### W3. Ablation Study on Segment Assignment Strategy
> We would like to kindly suggest that the reviewer first refer to our response to Q2-2 for a more comprehensive understanding. Our empirical results, shown in the table below, indicate that assigning every segment to every cross-attention layer is redundant; instead, we find that strategically distributing segments across layers is sufficient. In particular, allocating the most relevant segment to the final layers—where action prediction is most directly influenced—proves effective, offering a more refined and efficient design for the overall framework.
> |Strategy|# Segments per Layer|Object|Goal|
> |---|:---:|---|---|
> |Random|1|0.566|0.536|
> |Only Most Relevant|1|0.556|0.532|
> |Most-to-Least Relevant|1|0.570|0.544|
> |Least-to-Most Relevant (Ours)|1|0.578|0.546|
> |All|4|0.578|0.548|
>
> &nbsp;
> ### W4, Q3. Additional Details on DTW-based Behavioral Alignment
> Given two trajectories $[(V_1,L,s_1,a_1),…]$ and $[(V’_1,L’,s’_1,a’_1),…]$, we first apply the DTW algorithm to their respective action sequences $[a_1,a_2,…]$ and $[a’_1,a’_2,…]$. This optimization determines which action pairs $(a_i,a’_j)$ should be matched to achieve the best behavioral alignment, which yields a warping path consisting of index pairs $(i,j)$ that minimizes the $L_2$ distance-based cumulative matching cost. Consequently, the aligned segment pairs $(V_i,L,s_i,A_i)$ and $(V’_j,L’,s’_j,A’_j)$ are designated as positive pairs, whereas unaligned pairs are treated as negatives. By optimizing this contrastive objective, the retriever is trained to align segments that share behavioral patterns.
>
> &nbsp;
> ### Q1. Additional Details on Trajectory Storage
> Please refer to Reviewer TMNX’s W2-1.
>
> &nbsp;
> ### Q2-1. Ceiling on $K$
> The reviewer’s point is valid; our current implementation supports $K$ up to the number of cross-attention layers. One potential avenue for scaling $K$ further would be to distribute segments across the action head’s denoising iterations, though this remains a subject for further investigation.
>
> &nbsp;
> ### Q2-2. Inference Latency with Respect to K
> To clarify, when $K$ is smaller than the number of cross-attention layers, segments are replicated such that each layer is assigned a segment; at $K=1$, for instance, the most relevant segment is utilized by all layers (details provided in Reviewer TMNX’s W2-2). Combined with the backbone VLM being the primary bottleneck, our framework maintains a nearly constant inference latency regardless of $K$ (Figure 11).
>
> &nbsp;
> ### Q4. Potential Risks of In-context Adherence
> As further discussed in our Impact Statement, there is a risk that malicious or adversarial guidance could be injected into the context, potentially resulting in unsafe robotic behaviors. Therefore, we recognize that safeguarding the buffer against adversarial injections and establishing automated data-validation protocols are vital areas for future research.
>
> &nbsp;
> ### L1. Discussion on Limitations
> We acknowledge several limitations and will include a dedicated discussion in the revised manuscript: (1) there is an upper bound on $K$, and performance gains tend to saturate as $K$ increases; (2) despite substantial improvements over prior work, the success rate has yet to reach the levels observed in in-distribution scenarios; and (3) real-world robot experiments remain limited in terms of task diversity and trial counts.

---

> > ### Author Rebuttal · Reviewer_WFs2 · 2026-04-01
> >
> > Thank you for the thorough rebuttal. My concerns have been fully addressed. As promised, I am raising my score from 3 (weak reject) to 4 (weak accept).

---

### Official Review · Reviewer_L55d · 2026-03-12

**Soundness:** 3
**Presentation:** 3
**Significance:** 2
**Originality:** 3
**Overall Recommendation:** 4
**Confidence:** 3

**Summary:**

RA-VLA enables robotic policies to adapt to novel tasks without retraining by retrieving behaviorally-relevant expert demonstrations. It introduces action-aware contrastive retrieval to prioritize functional motion over visual similarity, and a contextual adherence loss ensuring actions are grounded in retrieved demonstrations rather than pre-trained biases. With a scalable design maintaining constant inference latency, RA-VLA achieves substantial gains on LIBERO and real-world tasks over baselines.

**Compliance With Llm Reviewing Policy:**

Affirmed.

**Final Justification:**

Given the solid new evidence on W2, I am willing to raise my score slightly, but the scope limitations prevent a stronger endorsement.

**Key Questions For Authors:**

1. Does RA-VLA support cross-embodiment retrieval, where demonstration segments come from a different robot morphology (or humans) than the target policy? If not, how would the framework need to be extended to handle embodiment gaps, such as different action spaces, kinematic structures, or visual perspectives?

2. You fixed the margin parameter m = 0.01 across all experiments without ablation. How sensitive is RA-VLA's performance to this hyperparameter? Does the optimal margin vary significantly across different tasks (e.g., LIBERO-Spatial vs. LIBERO-Long) or environments (simulation vs. real-world)?

3. It is claimed that latency remains constant regardless of K retrieved segments due to pre-computation. However, the retrieval itself still requires searching over the buffer B. What is the computational cost of this nearest-neighbor search as the buffer scales to hundreds or thousands of demonstrations, and how do you ensure this doesn't become a bottleneck?

4. The retriever uses cosine similarity on pooled embeddings. Have you explored more sophisticated similarity metrics?

**Limitations:**

yes

**Strengths And Weaknesses:**

### Strength

1. Careful experimental design with strict train/evaluation task separation and leave-one-out protocols.

2. Well-structured paper with clear problem motivation and illustrative failure cases.

3. The paper addresses a practically important problem: enabling robots to adapt to novel tasks without expensive retraining or weight updates.


### Weaknesses


1. The demonstration buffer contains trajectories from the same embodiment as the target policy. There is no testing of retrieving demonstrations from a different robot (e.g., using a human video demonstration, or a different robotic arm) and adapting them to the UR5e.

2. All experiments use GR00t N1.5 as the VLA backbone. The paper does not demonstrate that RA-VLA's improvements transfer to other architectures (e.g., OpenVLA, π0), leaving open whether benefits are architecture-specific or general.

3. All real-world tasks (Stack Box, Throw Trash, Close Drawer, Press Pedal) involve similar pick-and-place or simple articulation. The framework is not tested on tasks requiring fine-grained manipulation (e.g., inserting a peg, folding cloth, etc.) which may intuitively more sensitive to the trajectory-based retrieval as proposed in this paper.

4. The paper does not test how performance degrades with imperfect demonstrations. In real-world deployment, demonstration quality varies, but all experiments use clean expert trajectories.

---

> ### Author Rebuttal · Authors · 2026-03-31
>
> We sincerely thank the reviewer for incisive and insightful comments. We hope our response adequately addresses your concerns, and will incorporate these clarifications into the revised manuscript.
>
> &nbsp;
> ### W1, Q1. Towards Cross-Embodiment In-Context Adaptation
> We agree that cross-embodiment adaptation is a vital frontier for the field. However, we believe that achieving robust in-context adaptation within a single embodiment is a fundamental prerequisite—a challenge where existing ICIL methods still face significant hurdles. We expect the insights gained from our work to serve as an essential foundation for the eventual transition to more complex, cross-platform adaptation scenarios.
>
> To extend RA-VLA to cross-embodiment settings, it is crucial to unify action representations [1] to facilitate consistent action-sequence-based DTW alignment and effective in-context adaptation across diverse platforms. Furthermore, scaling diverse robotic datasets will allow the model to emulate diverse cross-embodiment adaptation scenarios during training, which will enable to internalize the embodiment gap in a data-driven manner.
> > [1] Zheng et al. Universal Actions for Enhanced Embodied Foundation Models. CVPR 2025.
>
> &nbsp;
> ### W2. Generalization to Other VLA Backbone
> To evaluate the generalizability of our framework across different VLA backbones, we adopt $\pi_{0.5}$ as the backbone and assess the in-context adaptability of the baselines on the LIBERO Spatial and Long suites. Since $\pi_{0.5}$ utilizes a Mixture-of-Transformer architecture, we pre-compute and store multimodal features from all VLM layers in the buffer.
>
> As shown in the table below, both action-aware retrieval and contextual adherence loss significantly enhance in-context adaptability of $\pi_{0.5}$, proving that their efficacy is not confined to a specific architecture.
> |Retriever|Method|Spatial|Long|
> |---|---|---|---|
> |Off-the-shelf|RICL-R|0.068|0.008|
> |Action-aware (Ours)|RICL-R|0.224|0.080|
> |Action-aware (Ours)|RA-VLA|0.366|0.178|
>
> &nbsp;
> ### W3. Evaluation on More Complex Tasks
> Given that training-free adaptation to novel tasks is inherently more challenging than in-distribution settings, we focused our evaluation on a set of representative tasks to validate this fundamental capability. While we agree that evaluating more intricate tasks would offer additional insights, we believe our results clearly demonstrate the distinct advantages of our method compared to prior work.
>
> &nbsp;
> ### W4. Adaptation using Imperfect Demonstrations
> For an experimental analysis regarding the impact of imperfect demonstrations, please refer to Reviewer TMNX’s L1-2.
>
> &nbsp;
> ### Q2. Sensitivity Analysis of Margin $m$
> We set the margin $m = 0.01$ for all experiments, motivated by the consistent training dynamics of our model. Specifically, we find that the in-distribution training loss $\mathcal{L}_{rel}$ consistently converges to a range of 0.05-0.07 across both LIBERO and UR5e settings; setting $m$ to roughly 10-30% of this converged loss value consistently yields optimal results.
>
> The following table illustrates the performance sensitivity to $m$: an excessively small margin fails to provide sufficient discriminative signals to distinguish contexts, whereas an overly large margin leads to training instability. Since $m$ can be calibrated using pre-deployment in-distribution loss, our framework remains practical for real-world deployment.
> |$m$|Spatial|Long|
> |---|---|---|
> |0.001|0.184|0.056|
> |0.005|0.314|0.124|
> |0.010|0.318|0.134|
> |0.020|0.310|0.136|
> |0.100|0.048|0.000|
>
> &nbsp;
> ### Q3. Retrieval Efficiency
> In RA-VLA, retrieval is performed via a matrix multiplication between the query ([1 x 512]) and the stored key ([N x 512]) matrices, followed by a Top-K operation. Notably, our benchmarks on an NVIDIA A100 GPU show a latency of just 0.00018s (~0.36% of the total inference latency) for $N = 10^7$, which ensures that the retrieval stage does not become a bottleneck as the buffer scales. This design can be further scaled, as ANN techniques can be seamlessly integrated to handle an even larger volume of demonstrations.
>
> &nbsp;
> ### Q4. Design Choices for Retriever
> Following recent retrieval literature [2-3], we adopt average pooling with cosine similarity—a widely adopted configuration for simplicity and efficacy. This setup has become an established standard in the field, as dot product-based measures are highly optimized for modern hardware and integrate seamlessly with ANN indexing techniques, ensuring superior computational scalability. While this setup provides a robust baseline, exploring more sophisticated similarity measures represents a valuable direction for future enhancement of the framework.
> > [2] Babakhin et al. Llama-Embed-Nemotron-8B: A Universal Text Embedding Model for Multilingual and Cross-Lingual Tasks. arXiv 2025.
> >
> > [3] Zhao et al. KaLM-Embedding-V2: Superior Training Techniques and Data Inspire A Versatile Embedding Model. ICLR 2026.

---

> > ### Author Rebuttal · Reviewer_L55d · 2026-04-06
> >
> > I thank the authors for their thorough response. The new backbone experiments (W2) demonstrating consistent improvements across architectures are convincing, and the margin sensitivity ablation (Q2) and retrieval latency analysis (Q3) adequately address those concerns.
> >
> > However, W1 and W3 remain open. Cross-embodiment retrieval is acknowledged as future work without empirical evidence, and the evaluation still lacks fine-grained manipulation tasks. These limit the generality claims of the framework. I will keep my original assesment.
> >
> > Given the solid new evidence on W2, I am willing to raise my score slightly.

---

> > > ### Author Response · Authors · 2026-04-08
> > >
> > > Once again, we deeply thank the reviewer’s insightful feedback, which has allowed us to strengthen the discussion on RA-VLA’s design, scalability, and applicability. We recognize that extending to cross-embodiment in-context adaptation is a significant research direction. Although the short rebuttal period precludes a rigorous experimental study, we would like to discuss how each of the two constitute phases of our framework—**expert segment retrieval** and **context-conditioned action generation**—can be effectively extended to cross-embodiment scenarios.
> > >
> > > &nbsp;
> > >
> > > ---
> > >
> > > ### 1. Retrieval in Cross-Embodiment Scenarios
> > > Our retrieval system is inherently capable of identifying relevant segments from cross-embodiment demonstrations. This is primarily achieved through our framework’s DTW-based alignment, which effectively mitigates potential temporal or phase shifts between different embodiments to identify behaviorally aligned expert segments. Such alignment is facilitated by representing actions through unified End-Effector (EEF) poses (e.g., [4]), ensuring a standardized representation across varying hardware platforms.
> > >
> > > ---
> > >
> > > ### 2. Action Generation (Adaptation) in Cross-Embodiment Scenarios
> > > We believe that *in-context adherence*—the core focus of our work—is fundamental to the effective utilization of expert cues from disparate embodiments. To further empower the VLA backbone in addressing both perceptual variances and intrinsic control discrepancies, we emphasize the importance of emulating diverse cross-embodiment adaptation scenarios during training. By scaling such robotic datasets, the model can internalize the embodiment gap in a data-driven manner, providing a robust foundation for cross-embodiment adaptation.
> > >
> > > ---
> > >
> > > > [4] Zheng et al. X-VLA: Soft-Prompted Transformer as Scalable Cross-Embodiment Vision-Language-Action Model. ICLR 2026.

---

### Official Review · Reviewer_TMNX · 2026-03-12

**Soundness:** 3
**Presentation:** 3
**Significance:** 3
**Originality:** 3
**Overall Recommendation:** 4
**Confidence:** 4

**Summary:**

This paper addresses the adaptation bottleneck of In-Context Imitation Learning (ICIL) for Vision-Language-Action (VLA) models at test time. To overcome superficial retrieval, behavioral inertia, and high inference latency, the authors propose RA-VLA. This framework features a DTW-based action-aware contrastive retriever, a contextual adherence loss ($L_{adhere}$) to break behavioral inertia, and a decoupled pairwise grounding architecture. Experiments in simulation and on a real-world robot demonstrate improved success rates on novel tasks while maintaining very low and constant inference latency.

**Compliance With Llm Reviewing Policy:**

Affirmed.

**Final Justification:**

My concerns have been addressed. I maintain my current score.

**Key Questions For Authors:**

Could you clarify the setup for Table 3? The text states it ablates $L_{adhere}$, but the label says "w/o $L_{align}$". Is this a typo?

How exactly are the "functional segments" in the buffer partitioned? This directly impacts the retriever's input.

In the pairwise grounding architecture, how are the $K$ retrieved segments precisely mapped and distributed across the cross-attention layers?

Could you provide direct quantitative metrics for the retrieval module itself (e.g., alignment accuracy vs. DTW ground truth), or include a non-learned DTW retrieval baseline for comparison?

**Limitations:**

The discussion on malicious data injection in the impact statement is pertinent. However, system-level limitations are under-discussed: e.g., what are the memory and pre-computation scaling costs of maintaining the offline context buffer? Additionally, how robust is the system if the buffer contains only sub-optimal demonstrations?

**Strengths And Weaknesses:**

Strengths:

Addresses a practical pain point: Training-free VLA test-time adaptation is highly practical. The paper clearly breaks down the latency and context-ignoring issues in current ICIL methods.

Clever architecture: Offline caching of segment encoding decouples retrieval from cross-attention execution, drastically improving latency scaling (Fig 5).

Solid pipeline: The design forms a logical closure—from behavior-aligned retrieval to context-grounded action generation—yielding tangible improvements in physical deployments.

Weaknesses:

Ambiguity in ablation study: The text claims Table 3 ablates the contextual adherence loss ($L_{adhere}$), but the table label reads "w/o $L_{align}$". This minor discrepancy creates confusion regarding the specific contributions of the core modules.

The text does not clearly explain how trajectories are partitioned into "functional segments" (fixed length, overlapping, or action-heuristic?). Additionally, the exact mapping rule when the number of retrieved segments $K$ does not match the number of cross-attention layers is unspecified.

Baselines could be expanded: Including a simpler retrieval baseline (e.g., direct DTW nearest-neighbor without the learned contrastive encoder) would better isolate and prove the value of the proposed retriever.

---

> ### Author Rebuttal · Authors · 2026-03-31
>
> We appreciate your thorough and insightful comments. We hope the following clarifications resolve your queries, and we welcome any further questions.
>
> &nbsp;
> ### W1, Q1. Notational Typo in Table 3
> We apologize for the confusion; $L_{align}$ in Table 3 is a typo and should be corrected to $L_{adhere}$ to maintain consistency with the main text. These results demonstrate that the contextual adherence loss ensures the model follows the expert guidance in the context, enabling it to effectively adapt to unseen tasks.
>
> &nbsp;
> ### W2-1, Q2. Additional Details on Trajectory Partitioning
> We will incorporate a clear description of the trajectory processing procedure into the main text to ensure a precise grasp of our framework. To clarify, the trajectory is partitioned using a sliding window of length $h = 16$ with a stride of $s = 8$. Specifically, a trajectory $[(V_1, L, s_1, a_1), ((V_2, L, s_2, a_2), …]$ is decomposed into segments such as $(V_1, L, s_1, A_1)$, $(V_9, L, s_9, A_9)$, and so on, where $A_t$ represents the action chunk $[a_t, …, a_{t+15}]$. Each segment $(V', L', s', A')$ is then stored in a buffer as a key-value pair: the key $z' = R(V', L')$ is used for retrieval, while the corresponding value $(H' = f_{\phi}(V', L'), s', A')$ provides the necessary guidance for adapting to new tasks.
>
> &nbsp;
> ### W2-2, Q3. Additional Details on Segment Assignments
> We manually designated the assignment of retrieved segments across the eight cross-attention layers to ensure a gradual and balanced distribution. For instance, when $K = 2$ (the number of retrieved segments), the first four cross-attention layers are assigned the 2nd most relevant segment, while the remaining four layers receive the 1st most relevant segment, following the segment assignment [2, 2, 2, 2, 1, 1, 1, 1]. The layer-wise segment assignments for other values of $K$ are provided in the table below. While our current implementation covers $K \le 8$, this strategy can be extended by dynamically adjusting assignments according to the denoising steps of flow matching.
>
> |K|Segment Assignment|
> |---|:---:|
> |1|[1, 1, 1, 1, 1, 1, 1, 1]|
> |2|[2, 2, 2, 2, 1, 1, 1, 1]|
> |3|[3, 3, 2, 2, 2, 1, 1, 1]|
> |4|[4, 4, 3, 3, 2, 2, 1, 1]|
> |5|[5, 4, 3, 3, 2, 2, 1, 1]|
> |6|[6, 5, 4, 3, 2, 2, 1, 1]|
> |7|[7, 6, 5, 4, 3, 2, 1, 1]|
> |8|[8, 7, 6, 5, 4, 3, 2, 1]|
>
> &nbsp;
> ### W3. Necessity of Retriever Model
> We wish to clarify that direct DTW-based retrieval is infeasible for test-time inference and evaluation. Since DTW alignment relies on action sequences to compute correspondence, it cannot be performed during inference where only observation-related data $(V_t, L, s_t)$ are accessible and the action sequence is unavailable. To address this, we leverage the DTW-aligned pairs, obtained through the ground-truth actions of the training demonstrations, as supervision labels to train the retriever. This allows the retriever to approximate the DTW-based retrieval using only available inputs $(V_t, L)$, bridging the information gap between training and testing. We will supplement the manuscript with a more comprehensive explanation of this rationale.
>
> &nbsp;
> ### L1-1. Overhead of Maintaining Context Buffer
> In practice, a single expert demonstration occupies ~30MB when stored in the context buffer, which is negligible compared to the overall memory footprint of the VLA model. Moreover, pre-computing the multimodal features for a single expert segment requires only ~0.025s on an A100 GPU, which can be further optimized through batch processing. Given that this is a one-time offline process, we believe the computational cost is highly reasonable and practical. Please refer to our response to Reviewer THMj’s W3-1 for a comprehensive discussion on the efficiency of the retrieval system.
>
> &nbsp;
> ### L1-2. Adaptation using Imperfect Demonstrations
> Since the buffer serves as the sole source for adapting to unseen tasks, the quality of expert demonstrations inherently dictates the overall performance. To investigate the impact of demonstration quality, we conducted real-world UR5e experiments on ‘Throw Trash’ and ‘Close Drawer’ using four imperfect demonstrations per task: two where the expert recovers from sub-goal failures to ultimately succeed, and two where the goal is reached through noisy movements.
>
> Across all baselines, we utilize our action-aware retriever and report the average success rate over 12 trials per task (same setup as in Table 2). As shown in the table below, RA-VLA demonstrates a notable resilience to imperfect demonstrations, outperforming RICL-R. This robustness is driven by a self-correction mechanism: although the model may initially mirror erroneous patterns, it eventually succeed by identifying and aligning with successful segments.
>
> | Method | Throw Trash (Pick Trash) | Close Drawer (Touch Drawer) |
> | --- | :---: | :---: |
> | RICL-R | 0.167 (0.333) | 0.250 (0.417) |
> | RA-VLA | 0.500 (0.583) | 0.583 (0.667) |

---

> > ### Author Rebuttal · Reviewer_TMNX · 2026-04-03
> >
> > Thank you for the detailed clarifications. My concerns have been addressed. I maintain my current score.

---

### Official Review · Reviewer_THMj · 2026-03-15

**Soundness:** 3
**Presentation:** 3
**Significance:** 3
**Originality:** 3
**Overall Recommendation:** 5
**Confidence:** 4

**Summary:**

This paper proposes a retrieval-augmented vision-language-action framework for test-time adaptation to unseen manipulation tasks. The method combines an action-aware retriever, trained with DTW-based contrastive alignment, with a grounded action generation objective that encourages the policy to rely on retrieved expert context rather than defaulting to its pre-trained prior. The paper reports improvements over vanilla VLA and prior in-context imitation baselines on LIBERO and on a real-world UR5e setup, while also arguing that the proposed architecture avoids the inference-time scaling issues of prompt-concatenation approaches. The work aims to present an important concept for making VLA systems more adaptable at test time, and the authors analyze the concept through retrieval, contextual sensitivity, and latency studies.

**Compliance With Llm Reviewing Policy:**

Affirmed.

**Final Justification:**

The rebuttal has addressed my concerns.

**Key Questions For Authors:**

Please address weaknesses in the review above.

**Limitations:**

Yes

**Strengths And Weaknesses:**

Strength
1. The paper addresses an important problem, namely how to adapt VLA policies to unseen tasks at test time without weight updates, and the empirical gains over the reported baselines are meaningful.
2. The combination of behavior-aligned retrieval and an explicit contextual adherence objective is sensible, and the contextual sensitivity analysis helps support the claim that the model is using retrieved context rather than only memorized priors.
3. The inference-scalability discussion is useful, and the near-constant latency claim is a practical strength for robotics deployment.


Weakness:
1. At the same time, the novelty should be discussed more carefully relative to prior in-context imitation learning and older one-shot imitation formulations that first encode demonstration context and then condition action prediction on the current observation plus context.
2. The discussion of related work on efficient inference is incomplete. In particular, there are state-space-model-based alternatives such as RoboSSM that are also relevant when arguing low inference cost on manipulation benchmarks.
3. The paper would benefit from a clearer discussion of the systems question behind retrieval at scale, namely how context is stored, indexed, and retrieved with low memory and I/O overhead in realistic deployments. It would also be good to discuss if the model can still be used as VLA by evaluating in-distribution performance on LIBERO or training tasks.

[1] Yoo, Youngju, Jiaheng Hu, Yifeng Zhu, Bo Liu, Qiang Liu, Roberto Martín-Martín, and Peter Stone. "RoboSSM: Scalable In-context Imitation Learning via State-Space Models." arXiv preprint arXiv:2509.19658 (2025).
[2] Duan, Yan, Marcin Andrychowicz, Bradly Stadie, OpenAI Jonathan Ho, Jonas Schneider, Ilya Sutskever, Pieter Abbeel, and Wojciech Zaremba. "One-shot imitation learning." Advances in neural information processing systems 30 (2017).

---

> ### Author Rebuttal · Authors · 2026-03-31
>
> We sincerely thank the reviewer for the constructive feedback. In the following, we provide detailed responses to the questions and concerns raised.
>
> &nbsp;
>
> ### W1. Further Discussion on Related Literature
> We will expand our literature review to include one-shot imitation learning [1-4] and additional in-context imitation learning methods [5-7]. While the concept of learning from a single demonstration [1] has been extensively developed, subsequent approaches often necessitate test-time model updates [2], are intrinsically tied to non-RGB modalities (e.g., depth or point clouds) [3-5], or lack compatibility with recent VLM-based VLA models [6], which currently represent the SOTA in robotic manipulation.
>
> In addition to overcoming these constrains, our primary contribution lies in exposing and addressing superficial retrieval and *behavior inertia*—fundamental bottlenecks that prior works have failed to address. Regarding [7], its focus on action tokenization is orthogonal to our objective, representing a complementary advancement rather than a direct alternative.
>
> > [1] Duan et al. One-Shot Imitation Learning. NIPS 2017.
> >
> > [2] Finn et al. One-Shot Visual Imitation Learning via Meta-Learning. CoRL 2017.
> >
> > [3] Vitiello et al. One-Shot Imitation Learning: A Pose Estimation Perspective. CoRL 2023.
> >
> > [4] Biza et al. One-shot Imitation Learning via Interaction Warping. CoRL 2023.
> >
> > [5] Palo et al. Keypoint Action Tokens Enable In-Context Imitation Learning in Robotics. RSS 2024.
> >
> > [6] Yoo et al. RoboSSM: Scalable In-context Imitation Learning via State-Space Models. CoRL 2025.
> >
> > [7] Vuong et al. Action Tokenizer Matters in In-Context Imitation Learning. IROS 2025.
>
> &nbsp;
>
> ### W2. Efficiency Comparison with RoboSSM
> Efficiency comparison between RA-VLA (ours) and RoboSSM should account for their distinct designs. First, RA-VLA employs a retrieval-based approach that processes only the most relevant segment (one image), whereas RoboSSM ingests the entire demonstration sequence (hundreds of images), leading to a larger input scale. Second, by restricting VLM usage to the prefill stage (feature extraction) rather than decoding stage, RA-VLA benefits from highly parallelizable computations that are significantly accelerated using GPUs and FlashAttention kernels.
>
> Admittedly, an empirical comparison with RoboSSM would be ideal; however, we encountered technical challenges in deploying their official codebase within the short rebuttal period. Nevertheless, we expect our framework to be more effective as it leverages a VLA backbone (e.g., GR00T-N1.5) pretrained on massive robotic data, which provides a more robust foundation than SSM-based backbones (e.g., Longhorn).
>
> &nbsp;
>
> ### W3-1. Efficiency of Retrieval System in Deployment
> To illustrate the computational efficiency of our retrieval system, consider a typical expert demonstration consisting of 128 action steps. This trajectory is partitioned into 15 segments, each stored in a GPU-managed buffer with a 1KB retrieval key vector, a 2MB pre-computed multimodal feature tensor, and a 1KB action chunk tensor (for the detailed processing pipeline, see our response to Reviewer TMNX’s W2-1). Such memory requirements are practically insignificant given the scale of modern GPU memory.
>
> In our current implementation, retrieval is performed via a low-latency matrix multiplication between the query ([1 x 512]) and the stored keys ([N x 512]), followed by a Top-K operation. Notably, our benchmarks on an NVIDIA A100 GPU show a latency of just 0.00018s (0.36% of the total inference latency) for $N = 10^7$, which ensures that the retrieval stage does not become a bottleneck as the buffer scales. This design can be further scaled, as techniques like ANN indexing can be seamlessly integrated to handle an even larger volume of demonstrations.
>
> &nbsp;
>
> ### W3-2. Performance on In-Distribution Tasks
> The table below summarizes the in-distribution (ID) performance across three LIBERO suites (with Goal suite held-out), utilizing our action-aware retriever with a buffer populated by ID demonstrations. RA-VLA matches the performance of Vanilla VLA, validating its applicability to seen tasks as well. RAEA also shows comparable results; despite its poor context adherence, it benefits from the internalized task knowledge acquired during training. Conversely, RICL-R exhibits suboptimal performance because it employs a weighted sum of retrieved action chunks and model predictions, which inadvertently degrades execution precision for ID tasks.
>
> | Method | Spatial | Object | Long |
> | --- | --- | --- | --- |
> | Vanilla VLA | 0.942 | 0.992 | 0.898 |
> | RAEA | 0.950 | 0.994 | 0.880 |
> | RICL-R | 0.612 | 0.752 | 0.428 |
> | RA-VLA | 0.938 | 0.992 | 0.892 |

---

> > ### Author Rebuttal · Reviewer_THMj · 2026-04-03
> >
> > My concerns have been fully addressed.

---

### Decision · Program_Chairs · 2026-04-30

**Decision:**

Accept (regular)

**Comment:**

All the reviewers have advocated the paper's acceptance, with highlights on the well-motivated designs of the methodology, addressing important problem, and thorough evaluation protocol. While there were several concerns about lacking discussion on the prior work, missing details and analyses, most of them are resolved in the rebuttal phase. While there was a remaining concern on the lack of evaluation on cross-embodiment retrieval, I agree with the authors in that it is more of a future work direction. Therefore, I agree with the reviewers and recommend the acceptance of this paper.